



# Mineralization of organic matter in boreal lake sediments: Rates, pathways and nature of the fermenting substrates

François Clayer[1,3], Yves Gélinas[2,3], André Tessier[1], Charles Gobeil[1,3]

[1]INRS-ETE, Université du Québec, 490 rue de la Couronne, Québec (QC), Canada G1K 9A9
[2]Concordia University, Department of Chemistry and Biochemistry, 7141 Sherbrooke Street West, Montreal (QC), Canada H4B 1R6
[3]Geotop, Interuniversity research and training centre in geosciences, 201 Président-Kennedy Ave., Montréal (QC), Canada H2X 3Y7

*Correspondence to*: François Clayer (francois.clayer@niva.no)

**Abstract.** The complexity of organic matter (OM) degradation mechanisms represents a significant challenge for developing biogeochemical models to quantify the role of aquatic sediments in the climate system. The common representation of OM by carbohydrates formulated as $CH_2O$ in models comes with the assumption that its degradation by fermentation produces equimolar amounts of methane ($CH_4$) and dissolved inorganic carbon (DIC). To test the validity of this assumption, we modeled using reaction-transport equations vertical profiles of the concentration and isotopic composition ($\delta^{13}C$) of $CH_4$ and

DIC in the top 25 cm of the sediment column from two lake basins, one whose hypolimnion is perennially oxygenated and one with seasonal anoxia. Our results reveal that methanogenesis only occurs via hydrogenotrophy in both basins. Furthermore, we calculate, from $CH_4$ and DIC production rates associated with methanogenesis, that the fermenting OM has an average carbon oxidation state (COS) below $-0.9$. Modeling solute porewater profiles reported in the literature for four other seasonally anoxic lake basins also yields negative COS values. Collectively, the mean ($\pm$SD) COS value of $-1.4 \pm 0.3$ for all the seasonally

anoxic sites is much lower than the value of zero expected from carbohydrates fermentation. We conclude that carbohydrates do not adequately represent the fermenting OM and that the COS should be included in the formulation of OM fermentation in models applied to lake sediments. This study highlights the need to better characterize the labile OM undergoing mineralization to interpret present-day greenhouse gases cycling and predict its alteration under environmental changes.

## 1 Introduction

Significant proportions of atmospheric methane ($CH_4$) and carbon dioxide ($CO_2$), two powerful greenhouse gases, are thought to originate from freshwater lake sediments (Wuebbles and Hayhoe 2002; Bastviken et al., 2004; Turner et al., 2015), but large uncertainties remain concerning their contribution to the global $CO_2$ and $CH_4$ budgets (Saunois et al., 2016). The role of these waterbodies in the global carbon (C) budget has been acknowledged for more than a decade (Cole et al. 2007). Especially in the lake-rich boreal region, lakes are hotspots of $CO_2$ and $CH_4$ release (Hastie et al., 2018; Wallin et al., 2018) and intensive

sites of terrestrial C processing (Holgerson and Raymond, 2016; Staehr et al., 2012). Using high-resolution satellite imagery,



Verpoorter et al. (2014) estimated to about 27 million the number of lakes larger than 0.01 $km^2$ on Earth and reported that the highest lake concentration and surface area are found in boreal regions. Boreal lakes, which are typically small and shallow, are known to store large amounts of organic C, to warm up quickly, and to develop anoxic hypolimnia in the warm season (Schindler et al., 1996; Sabrekov et al., 2017). Owing to the great abundance of boreal lakes, their sensitivity to climate change

and foreseen important role in the global C cycle, there is a need to further develop process-based models to better quantify C processing reactions in these lakes and their alteration under warming (Saunois et al., 2016).

In aquatic environments, $CH_4$ is mainly produced (methanogenesis) in the sediment along with $CO_2$ at depths where most electron acceptors (EAs) are depleted (Conrad, 1999; Corbett et al., 2013). During its upward migration to the atmosphere, $CH_4$ is partly aerobically or anaerobically oxidized to $CO_2$ (methanotrophy) in the upper strata of the sediments and in the

water column (Bastviken et al., 2008; Raghoebarsing et al. 2006; Beal et al. 2009; Ettwig et al. 2010; Egger et al. 2015). The oxidation of organic matter (OM) by EAs such as $O_2$, $NO_3^-$, Fe(III), Mn(IV), $SO_4^{2-}$ and humic substances, as well as the partial fermentation of high molecular weight organic matter (HMW OM) into lower molecular weight organic matter (LMW OM) are also potential sources of $CO_2$ in the sedimentary environment (Corbett et al., 2015). Predicting fluxes of $CH_4$ and $CO_2$ from the aquatic sediments and water column to the atmosphere is challenging considering the various transport processes and

chemical and microbially-mediated reactions implicated and the complexity of natural OM which serves as substrate (Natchimuthu et al., 2017).

Process-based geochemical models taking into account both the numerous biogeochemical reactions involving C and transport processes are powerful tools able to interpret present-day sediment, porewater and water-column profiles of C species and offer a great potential to forecast changes in cycling of this element under variable environmental scenarios (Wang and Van

Cappellen 1996; Arndt et al., 2013; Paraska et al., 2014; Saunois et al., 2016). Nonetheless, the performance of these models depends on the correct formulation of the OM mineralization reactions, particularly in terms of the metabolizable organic compounds involved. Up to now, carbohydrates, represented as the simple chemical formula $CH_2O$ (or $C_6H_{12}O_6$), whose average carbon oxidation state (COS) is zero, are commonly assumed to be representative of the bulk of metabolizable OM, including the substrates involved in fermentation reactions (e.g., Arndt et al., 2013; Paraska et al., 2014; Arning et al., 2016

and references therein). The capacity of $CH_2O$ to represent adequately the ensemble of labile organic compounds is, nevertheless, becoming increasingly questioned in the literature given the variety and complexity of organic molecules present in the environment (Alperin et al., 1994; Berelson et al., 2005; Jørgensen and Parkes, 2010; Burdige and Komada, 2011; Clayer et al., 2016). Based on the observation that methanogenesis produced $CH_4$ three times faster than $CO_2$ in the sediments of a boreal, sporadically anoxic lake basin, Clayer et al. (2018) concluded that the fermenting OM had a markedly negative COS

value of −1.9. This COS value corresponds more closely to a mixture of fatty acids and fatty alcohols than to carbohydrates (e.g., $CH_2O$), which would have yielded equivalent $CH_4$ and $CO_2$ production rates. The low COS value of metabolizable OM in the sediment layer where methanogenesis occurred in this lake has been attributed to the nearly complete consumption of the most labile organic components (e.g., carbohydrates, proteins) during its downward transport through the water column and the upper sediment layers, thus leaving only material of lower lability such as fatty acids and fatty alcohols available for



methanogenesis. Such interpretation, however, must be validated by investigating other lakes before revising the formulation
of the fermenting OM used in diagenetic models in order to improve model predictions of C cycling, including greenhouse
gases production and emission from these environments.

In this study, centimeter-scale vertical porewater profiles of the concentrations and of the stable carbon isotope ratios ($\delta^{13}C$)
of $CH_4$ and dissolved inorganic carbon (DIC), as well as those of the concentrations of EAs were obtained in the hypolimnetic

sediments of two additional boreal lake basins showing contrasted $O_2$ dynamics: one whose hypolimnion remains perennially
oxygenated and the other whose hypolimnion becomes anoxic for several months annually. Reaction-transport equations are
used to quantify the rates of each OM mineralization pathway and estimate the COS of the substrates fermenting in the
sediments. Additional insight into the COS of the fermenting OM in lakes is also provided by applying these equations to
similar porewater solute concentration profiles gathered from the scientific literature or from our data repository.

**2 Materials and Methods**

**2.1 Sites and sample collection**

This study was carried out in two small, dimictic, oligotrophic and headwater lakes located within 50 km from Québec City,
Eastern Canada and having fully forested and uninhabited watersheds (Fig. 1). Lake Tantaré (47°04'N, 71°32'W) is part of
the Tantaré Ecological Reserve and has four basins connected by shallow channels and a total surface area of 1.1 $km^2$. Lake

Bédard (47°16'N, 71°07'W), lying in the protected Montmorency Forest, comprises only one small (0.05 $km^2$) basin. The
samples for this study were collected at the deepest sites of Lake Bédard (10 m) and of the westernmost basin of Lake Tantaré
(15 m), thereafter referred to as Basin A of Lake Tantaré to remain consistent with our previous studies (e.g., Couture et al.,
2008; Clayer et al., 2016). These two sampling sites were selected based on their contrasting $O_2$ regimes (Fig. 1): Lake Bédard
develops an anoxic hypolimnion early in the summer (D'arcy, 1993), whereas the hypolimnion of Lake Tantaré Basin A is

perennially oxygenated (Couture et al., 2008). The $O_2$ diffusion depth in the sediments of Lake Tantaré Basin A, as measured
with a microelectrode, does not exceed 4 mm (Couture et al., 2016).

Sediment porewater samples were acquired by *in situ* dialysis in October 2015 with peepers (Hesslein, 1976; Carignan et al.,
1985) deployed by divers within a 25-m² area at the deepest site of each lake basin. Bottom water $O_2$ concentrations were ~2.5
and < 0.1 mg $L^{-1}$ in Lake Tantaré Basin A and in Lake Bédard, respectively. The acrylic peepers comprised two columns of

4-mL cells, filled with ultrapure water, and covered by a 0.2-µm Gelman HT-200 polysulfone membrane, which allowed
porewater sampling from about 23–25 cm below the sediment-water interface (SWI) to 5 cm above this interface (thereafter
referred to as overlying water) at a 1-cm depth resolution. Oxygen was removed from the peepers prior to their deployment,
as described by Laforte et al. (2005). Four peepers were left in the sediments of each lake basin for at least 15 d, i.e., a longer
time period than that required for solute concentrations in the peeper cells to reach equilibrium with those in the porewater (5–

10 d; Hesslein, 1976; Carignan et al., 1985). At least three independent porewater profiles of pH, of the concentrations of $CH_4$,
DIC, acetate, $NO_3^-$, $SO_4^{2-}$, Fe and Mn, and of the $\delta^{13}C$ of $CH_4$ and DIC were generated for the two sampling sites. In Lake



Bédard, samples were also collected to determine three porewater profiles of sulfide concentrations ($\Sigma S(-II)$). After peeper

retrieval, samples (0.9–1.9 mL) for $CH_4$ and DIC concentrations and $\delta^{13}C$ measurements were collected within 5 minutes from

the peeper cells with He-purged polypropylene syringes. They were injected through rubber septa into He-purged 3.85-mL

exetainers (Labco Limited), after removal of a volume equivalent to that of the collected porewater. The exetainers were

preacidified with 40–80 µL of HCl 1N to reach a final pH ≤ 2. The protocols used to collect and preserve water samples for

the other solutes are given by Laforte et al. (2005).

### 2.2 Analyses

Concentrations and carbon isotopic composition of $CH_4$ and DIC were measured as described by Clayer et al. (2018). Briefly,

the concentrations were analyzed within 24 h of peeper retrieval by gas chromatography with a precision better than 4 % and

detection limits (DL) of 2 µM and 10 µM for $CH_4$ and DIC, respectively. The $^{13}C/^{12}C$ abundance ratios of $CH_4$ and $CO_2$ were

determined by Mass Spectrometry with a precision of ± 0.2 ‰ when 25 µmol of an equimolar mixture of $CH_4$ and $CO_2$ was

injected, and results are reported as:

$$\delta^{13}C = 1000 \left( \frac{\left( \frac{^{13}C_{solute}}{^{12}C_{solute}} \right)_{sample}}{\left( \frac{^{13}C}{^{12}C} \right)_{standard}} - 1 \right) \tag{1}$$

where the subscript solute stands for $CH_4$ or DIC and the reference standard is Vienna Pee Dee Belemnite (VPDB). Acetate

concentration was determined by ion chromatography (DL of 1.4 µM) and those of Fe, Mn, $NO_3^-$, $SO_4^{2-}$ and $\Sigma S(-II)$, as given

by Laforte et al. (2005).

### 2.3 Modeling of porewater solutes and the reaction network

The computer program WHAM 6 (Tipping, 2002) was used, as described by Clayer et al. (2016), to calculate the speciation

of porewater cations and anions. The solute activities thus obtained, together with solubility products ($K_s$), were used to

calculate saturation index values (SI = log IAP/$K_s$, where IAP is the ion activity product).

The following one-dimensional mass-conservation equation (Boudreau, 1997):

$$\frac{\partial}{\partial x}\left( \varphi D_s \frac{\partial [solute]}{\partial x} \right) + \varphi \alpha_{Irrigation}([solute]_{tube} - [solute]) + R_{net}^{solute} = 0 \tag{2}$$

was used to model the porewater profiles of $CH_4$, DIC, $O_2$, Fe and $SO_4^{2-}$, assuming steady state and negligible solute transport

by bioturbation and advection (Clayer et al., 2016). In this equation, [solute] and $[solute]_{tube}$ denote a solute concentration

in the porewater and in the animal tubes (assumed to be identical to that in the overlying water), respectively, x is depth

(positive downward), $\varphi$ is porosity, $D_s$ is the solute effective diffusion coefficient in sediments, $\alpha_{Irrigation}$ is the bioirrigation

coefficient, and $R_{net}^{solute}$ (in mol cm$^{-3}$ of wet sediment s$^{-1}$) is the solute net production rate (or consumption rate if $R_{net}^{solute}$ is

negative). $D_s$ was assumed to be $\varphi^2 D_w$ (Ullman and Aller, 1982), where $D_w$ is the solute tracer diffusion coefficient in water.





The values of $D_w$, corrected for in situ temperature (Clayer et al., 2018), were $9.5 \times 10^{-6}$ cm$^2$ s$^{-1}$, $6.01 \times 10^{-6}$ cm$^2$ s$^{-1}$ and $1.12 \times 10^{-5}$ cm$^2$ s$^{-1}$ for CH$_4$, HCO$_3^-$ and CO$_2$, respectively. The values of $\alpha_{Irrigation}$ in Lake Tantaré Basin A were calculated as in

Clayer et al. (2016), based on an inventory of benthic animals (Hare et al., 1994), and were assumed to be 0 in Lake Bédard since its bottom water was anoxic (Fig. 1).

The $R_{net}^{solute}$ values were determined from the average (n = 3 or 4) solute concentration profiles by numerically solving Eq. (2) with the computer code PROFILE (Berg et al., 1998). The boundary conditions were the solute concentrations at the top and at the base of the porewater profiles. In situ porewater O$_2$ profiles were not measured in Lake Tantaré Basin A. For modeling

this solute with PROFILE, we assumed that the [O$_2$] in the overlying water was identical to that measured in the lake bottom water and equal to 0 below 0.5 cm (based on O$_2$ penetration depth; Couture et al., 2016). This procedure provides a rough estimate of $R_{net}^{O_2}$ at the same vertical resolution as for the other solutes. The code PROFILE yields a discontinuous profile of discrete $R_{net}^{solute}$ values over depth intervals (zones) which are objectively selected by using the least square criterion and statistical F-testing (Berg et al., 1998). The fluxes of solute transport across the SWI due to diffusion and bioirrigation are also

estimated by PROFILE. In order to estimate the variability in $R_{net}^{solute}$ related to heterogeneity within the 25-m$^2$ sampling area, additional $R_{net}^{solute}$ values were obtained by modeling the average profiles whose values were increased or decreased by one standard deviation. This variability generally ranges between 2 and 10 fmol cm$^{-3}$ s$^{-1}$.

The main reactions retained in this study to describe carbon cycling in the sediments of the two lake basins are shown in Table 1. Once oxidants are depleted, fermentation of metabolizable OM (r1) can yield acetate, CO$_2$ and H$_2$. The partial degradation

of high molecular weight OM (HMW OM) into lower molecular weight OM (LMW OM) can also produce CO$_2$ (r2, Corbett et al., 2013; Corbett et al., 2015). Acetoclasty (r3) and hydrogenotrophy (r4) yield CH$_4$. Moreover, CH$_4$ (r5) and OM (r6) can be oxidized to CO$_2$ when electron acceptors such as O$_2$, Fe(III) and SO$_4^{2-}$ are present. Note that the electron acceptors (EAs) NO$_3^-$ and Mn oxyhydroxides can be neglected in these two lake basins (Feyte et al., 2012; Clayer et al., 2016) as well as the precipitation of metal carbonates whose saturation index values are negative (SI ≤ −1.5) except for siderite (r7) in Lake Bédard

(SI = 0.0 to 0.7). Lastly, sulfide oxidation by iron oxides (r8), which can be a source of SO$_4^{2-}$ and H$_2$ (Holmkvist et al., 2011; Clayer et al., 2018), is also considered.

From Table 1, the net rate of CH$_4$ production, $R_{net}^{CH_4}$, in the sediments is:

$$R_{net}^{CH_4} = R_3 + R_4 - R_5 \qquad (3)$$

where $R_3$ and $R_4$ are the rates of acetoclastic (r3) and hydrogenotrophic (r4) production of CH$_4$, respectively, and $R_5$ is the rate of DIC production due to CH$_4$ oxidation (r5). The net rate of DIC production, $R_{net}^{DIC}$, can be expressed as:

$$R_{net}^{DIC} = R_1 + R_2 + R_3 - R_4 + R_5 + R_6 - R_7 \qquad (4)$$

where $R_1$, $R_2$ and $R_6$ are the rates of DIC production due to complete fermentation of labile OM (r1), partial fermentation of HMW OM (r2) and OM oxidation (r6), respectively, and $R_7$ is the rate of DIC removal by siderite precipitation (r7). It can also be written that:





$$R_{net}^{Ox} = -2R_5 - R_6 \qquad (5)$$

where $R_{net}^{Ox}$ is the net reaction rate of all the oxidants ($O_2$, Fe(III) and $SO_4^{2-}$) consumption. For simplicity, $R_{net}^{Ox}$ is expressed in equivalent moles of $O_2$ consumption rate, taking into account that $SO_4^{2-}$ and Fe(III) have twice and one quarter the oxidizing

capacity of $O_2$, respectively. In practice, the value of $R_{net}^{Ox}$ was calculated by adding those of $R_{net}^{O_2}$, $\frac{1}{4}R_{net}^{Fe(III)}$ and $2R_{net}^{SO_4^{2-}}$ where $R_{net}^{O_2}$, $R_{net}^{Fe(III)}$ and $R_{net}^{SO_4^{2-}}$ were estimated with PROFILE. In this calculation, we assumed that all dissolved Fe is in the form of Fe(II), and that the rate of Fe(II) consumption through reactions r7 is negligible compared to those associated with reactions r5 and r6. Under these conditions, $R_{net}^{Fe(III)} = -R_{net}^{Fe}$.

## 2.4 Modeling of the $\delta^{13}C$ profiles

The $\delta^{13}C$ profiles of $CH_4$ ($\delta^{13}C$-$CH_4$) and DIC ($\delta^{13}C$-DIC) were simulated with a modified version of Eq. 1 (Clayer et al., 2018):

$$\delta^{13}C = 1000 \left( \frac{\left(\frac{[^{13}C]}{[C]}\right)_{sample}}{\left(\frac{^{13}C}{^{12}C}\right)_{standard}} - 1 \right) \qquad (6)$$

where [C] is the total $CH_4$ or DIC concentration ($[^{12}C]$ can be replaced by [C] since ~99% of C is $^{12}C$), and $[^{13}C]$ is the isotopically heavy $CH_4$ or DIC concentration. Equation 6 allows calculating a $\delta^{13}C$ profile once the depth distributions of $[^{13}C]$ and [C] are known. This information is obtained by solving the mass-conservation equations of C and $^{13}C$ for $CH_4$ and DIC.

The one-dimensional mass-conservation of [C] is given by Eq. 2 where [solute] is replaced by [C], whereas that for $[^{13}C]$ is the following modified version of Eq. 2 (Clayer et al., 2018):

$$\frac{\partial}{\partial x}\left( \varphi \frac{D_s}{f} \frac{\partial [^{13}C]}{\partial x} \right) + \varphi\alpha_{Irrigation}([^{13}C]_{tube} - [^{13}C]) + \sum_{i=1}^{5} \frac{R_i}{\alpha_i}\left( \frac{\delta^{13}C_i^{reactant}}{1000} + 1 \right)\left( \frac{^{13}C}{^{12}C} \right)_{standard} = 0 \qquad (7)$$

where f, the molecular diffusivity ratio, is the diffusion coefficient of the regular solute divided by that of the isotopically heavy solute, $\alpha_i$ is the isotope fractionation factor in reaction $r_i$, and $\delta^{13}C_i^{reactant}$ is the $\delta^{13}C$ of the reactant leading to the formation of the solute ($CH_4$ or DIC) in reaction $r_i$. Input and boundary conditions used to numerically solve Eqs 2 and 7 for

[C] and $[^{13}C]$, respectively, via the bvp5c function of MATLAB® are described in section 3.4 and in section S2 of the Supporting Information (SI).

The goodness of fit of the model was assessed with the norm of residuals ($N_{res}$):

$$N_{res} = \sqrt{\sum_{x=0.5}^{22.5} (\delta^{13}C_m - \delta^{13}C_s)^2} \qquad (8)$$





where $\delta^{13}C_m$ and $\delta^{13}C_s$ are the measured and simulated $\delta^{13}C$ values, respectively. The norm of residuals ($N_{res}$) varies between 0 and infinity with smaller numbers indicating better fits.

**2.5 Data treatment of other data sets**

To better assess the COS of the fermenting OM in lakes, relevant sets of porewater concentration profiles (CH$_4$, DIC, EAs, Ca) available from the literature or from our data repository have been modeled with the code PROFILE, as described in section 2.3, to extract their $R_{net}^{CH_4}$, $R_{net}^{DIC}$ and $R_{net}^{Ox}$ profiles. These porewater datasets, described in section S3 of the SI, had been generated by sampling porewater in the hypolimnetic sediments of: i) Lake Bédard and Basin A of Lake Tantaré, at other dates than for this study (Clayer et al, 2016); ii) Basin B of Lake Tantaré (adjacent to Basin A; Fig 1), on four occasions (Clayer et al., 2016; 2018); iii) Williams Bay of Jacks Lake (44°41' N, 78°02' W), located in Ontario, Canada, on the edge of the Canadian Shield (Carignan and Lean 1991); iv) the southern basin of the alpine Lake Lugano (46°00'N, 3°30'E) located in Switzerland, on two occasions (Lazzaretti-Ulmer and Hanselmann 1999). All lake basins, except Basin A of Lake Tantaré develop an anoxic hypolimnion.

**3 Results**

**3.1 Solute concentration profiles**

Differences among the replicate profiles of CH$_4$, DIC, SO$_4^{2-}$, $\Sigma$S(−II) and Fe (Fig. 2) at the two sampling sites are generally small (except perhaps those of SO$_4^{2-}$ in Lake Bédard) and should be mainly ascribed to spatial variability within the 25-m$^2$ sampling area. Indeed, the main vertical variations in the profiles are defined by several data points without the sharp discontinuities expected from sampling and handling artifacts. Note that the acetate concentrations, which were consistently low (< 2 µM), are not shown.

The low Fe (< 5 µM; Fig. 2f) and CH$_4$ (< 2 µM; Fig. 2a) concentrations as well as the relatively high SO$_4^{2-}$ concentrations (36 ± 2.1 µM; Fig. 2e) in the sediment overlying water of Lake Tantaré Basin A are all consistent with the [O$_2$] (~2.5 mg L$^{-1}$) measured in the bottom water and are indicative of oxic conditions at the sediment surface. The sharp Fe gradients near the SWI indicate an intense recycling of Fe oxyhydroxides (Fig. 2f; Clayer et al., 2016) and the concave-down curvatures in the SO$_4^{2-}$ profiles (Fig. 2e) reveal SO$_4^{2-}$ reduction near the SWI. In contrast to Lake Tantaré Basin A, high Fe (> 200 µM), measurable CH$_4$ (> 200 µM) low SO$_4^{2-}$ (2.7 ± 1.4 µM) and detectable $\Sigma$S(−II) concentrations in the overlying waters of Lake Bédard (Fig. 2i, m and n) are consistent with anoxic conditions at the sediment surface. The absence of a sharp Fe gradient at the SWI in Lake Bédard suggests that Fe oxyhydroxides were not recycled in these sediments when porewater sampling occurred.

In the two lake basins, SO$_4^{2-}$ concentrations reach a minimum between the SWI and 5 cm depth (Fig. 2e and m), and increase below these depths. Alongside, all Fe profiles show a slight increase downward (Fig. 2f and n) indicating that solid Fe(III) is





reduced to produce dissolved Fe. In Lake Bédard, the $\Sigma S(-II)$ concentrations decrease from the SWI to ~10 cm depth and remain relatively constant below that depth at $0.08 \pm 0.06$ µM for two of the profiles and at $0.71 \pm 0.18$ µM for the other one

(grey filled triangles in Fig. 2n).

The concentrations of $CH_4$ (< 1.5 mM; Fig. 2a and i) are well below saturation at 4°C and *in situ* pressure (4.4–5.5 mM; Duan and Mao, 2006), implying that $CH_4$ ebullition is a negligible $CH_4$ transport process. The $CH_4$ values increases from < 2 µM in the overlying water to 0.18–0.20 mM at the base of the Lake Tantaré Basin A profiles (Fig. 2a), and from 0.2–0.5 mM to 1.0–1.4 mM in those of Lake Bédard (Fig. 2i). The three $CH_4$ profiles from Lake Tantaré Basin A (Fig. 2a) show a modest concave-

up curvature in their upper part, close to the SWI, indicative of a net $CH_4$ consumption, and a convex-up curvature in their lower part, typical of a net $CH_4$ production. Such trends, however, are not observed in Lake Bédard sediments. The $CH_4$ profiles from this lake exhibit a convex-up curvature over the whole sediment column, although more pronounced in its upper part (Fig. 2i).

The DIC concentrations consistently increase from 0.27–0.32 mM and 1.2–1.5 mM in the sediment overlying water to 0.76–

0.83 mM and 3.5–4.3 mM at the bottom of the profiles in Lake Tantaré Basin A and Lake Bédard, respectively (Fig. 2c and k). All DIC profiles show a similar shape with a slight concave-up curvature in their lower segment and a convex-up curvature in their upper portions.

### 3.2 Modeled $CH_4$ and DIC concentration profiles

The modeled [$CH_4$] and DIC profiles accurately fit the average (n = 3 or 4) data points ($r^2 > 0.996$ and $r^2 > 0.998$ for $CH_4$ and

DIC, respectively; Fig. 2g,h,o and p). The $R_{net}^{CH_4}$ profiles reveal three zones in each lake basin numbered $Z_1$, $Z_2$ and $Z_3$ from the sediment surface whose boundaries match those defined by the $R_{net}^{DIC}$ profiles. For Lake Tantaré Basin A, $Z_1$ corresponds to a net $CH_4$ consumption and $Z_2$ and $Z_3$ to net $CH_4$ production, with the highest rate in $Z_2$ (Fig. 2g). In contrast, the three zones in Lake Bédard show net $CH_4$ production with the highest rate in $Z_1$ and the lowest in $Z_3$ (Fig. 2o). The $R_{net}^{DIC}$ profiles in both lake basins show a zone of net DIC consumption below two zones of net DIC production with the highest rate values in the $Z_1$ and

$Z_2$ for Lake Tantaré Basin A and Lake Bédard, respectively.

The $R_{net}^{CH_4}$ and $R_{net}^{DIC}$ profiles displayed in Figure 2 are, among all the possible solutions, the ones that give the simplest rate profile while providing a satisfying explanation of the averaged solute concentration profile as determined by statistical F-testing implemented in the code PROFILE (P value $\leq 0.001$ except for the $R_{net}^{DIC}$ profile in Lake Bédard whose P value is $\leq$ 0.005). As an additional check of the robustness of the depth distribution of $R_{net}^{CH_4}$ and $R_{net}^{DIC}$ provided by PROFILE, we used

another inverse model, i.e., Rate Estimation from Concentrations (REC; Lettmann et al., 2012) to model the average $CH_4$ and DIC profiles. Note that the statistical method, implemented in REC to objectively select the depth distribution of the net reaction rates, i.e., the Tikhonov regularization technique, differs from that of PROFILE. Figure S1 (SI) shows that the two codes predicted mutually consistent $R_{net}^{CH_4}$ and $R_{net}^{DIC}$ profiles, with rate values of similar magnitude. PROFILE was also used in this study to estimate $R_{net}^{SO_4^{2-}}$, $R_{net}^{Fe}$ and $R_{net}^{O2}$ in order to calculate the value of $R_{net}^{Ox}$ in each zone at both sampling sites (see





section 2.3 for details). The modeled $[SO_4^{2-}]$ and $[Fe]$ profiles are not shown but, again, they accurately fit the data points ($r^2 > 0.983$). As expected from the contrasting $O_2$ regimes of the two lake basins, $R_{net}^{Ox}$ values for Lake Tantaré Basin A were one to two orders of magnitude higher than those for Lake Bédard. Note that $R_{net}^{O2}$ was by far the highest contributor to the value of $R_{net}^{Ox}$ in Lake Tantaré Basin A with values of −290 and −72 fmol cm$^{-3}$ s$^{-1}$ in the $Z_1$ and $Z_2$, respectively. The values of $R_{net}^{CH_4}$, $R_{net}^{DIC}$ and $R_{net}^{Ox}$ estimated in each zone of each lake basins are reported in Table 2.

### 3.3 The δ¹³C profiles

The δ¹³C-DIC values increase from −28.2 ± 0.4 ‰ and −17.2 ± 0.7 ‰ in the overlying water to −5.1 ± 1.0 ‰ and 3.6 ± 1.7 ‰ at the base of the profiles in Lake Tantaré Basin A and Lake Bédard, respectively (Fig. 2d and l). Similarly, the δ¹³C-CH₄ values in Lake Bédard increase steadily from −82.5 ± 3.3 ‰ in the overlying water to −74.0 ± 1.5 ‰ at 24.5 cm depth (Fig. 2j). Regarding Lake Tantaré Basin A, the CH₄ concentrations above 1.5 cm depth were too low for their ¹³C/¹²C ratio to be

determined. Starting at 1.5 cm depth, the δ¹³C-CH₄ values first decrease from −91.1 ± 11.1 ‰ to −107.0 ± 6.8 ‰ at 2.5 cm depth and then increase progressively to −83.5 ± 1.6 ‰ at the base of the profiles (Fig. 2b). Note that a shift toward more positive δ¹³C-CH₄ values upward, generally attributed to the oxidation of CH₄ (Chanton et al., 1997; Norði et al., 2013), is only observed in the profiles of Lake Tantaré Basin A (Fig. 2b).

As shown in Fig. S2 (SI), the isotopic signatures of nearly all samples from the two lake basins fall within the ranges reported

for hydrogenotrophic methanogenesis, i.e., CO₂ reduction, in a δ¹³C-CO₂ *vs* δ¹³C-CH₄ graph similar to that proposed by Whiticar (1999). Indeed, the values of δ¹³C-CH₄ which are lower than -70 ‰ over the whole profiles in the two lake basins, and the large difference (67 to 92 ‰) between the δ¹³C of gaseous CO₂ (δ¹³C-CO₂) and δ¹³C-CH₄, strongly contrast with the typical δ¹³C-CH₄ values (−68 to −50 ‰) and with the difference between δ¹³C-CO₂ and δ¹³C-CH₄ (39 to 58 ‰) reported for acetoclasty (Whiticar, 1999). The δ¹³C results reported previously for another basin of Lake Tantaré (Basin B; Clayer et al.,

2018) show also in the hydrogenotrophy domain in Fig. S2.

### 3.4 Modeled δ¹³C profiles

In order to model the δ¹³C profiles with Eq. 6, accurate profiles of $[C]$ and $[^{13}C]$ need first to be determined by numerically solving Eqs. 2 and 7, respectively. The modeled profiles of $[CH_4]$ and DIC obtained with Eq. 2 replicated well the measured profiles of these two solutes when the depth distributions of $R_{net}^{CH_4}$ or $R_{net}^{DIC}$ provided by PROFILE (Table 2) and those of $D_s$,

$\alpha_{Irrigation}$ and $\varphi$ were used as inputs in Eq. 2, and when measured CH₄ or DIC concentrations at the top and bottom of the profiles were imposed as boundary conditions. Getting a truthful profile of $[^{13}C]$ with Eq. 7 requires, however, accurate values of $\delta^{13}C_i^{reactant}$, $\alpha_i$, and $R_i$ for each of the reactions given in Table 1, and of f for both CH₄ (f-CH₄) and DIC (f-DIC). The multi-step procedure followed to obtain the best $[^{13}C]$ profiles for CH₄ and DIC is described in section S2 (SI). This modeling exercise revealed that $R_3 = 0$ for all the zones in the sediments of both lake basins, thus confirming that practically all CH₄ is

produced through hydrogenotrophy, as inferred above from the δ¹³C values.



The best fits between the simulated and measured $\delta^{13}C$ profiles of $CH_4$ and DIC for Lake Tantaré Basin A and Lake Bédard (red lines in Fig. 3) were obtained with the f, $\alpha_i$ and $R_i$ values displayed in Table 3. The optimal $\alpha_i$ and f values were within the ranges reported in the literature for both lake basins, except for the lower-than-expected value of $\alpha_2$ (0.984) in the $Z_2$ of Lake Bédard. Note that $\alpha_3$ is not given in Table 3 since the modeling of the $\delta^{13}C$ profiles of $CH_4$ and DIC indicates that $R_3 = 0$

(see section S2.2.2.1 in the SI). Optimal values for $\alpha_4$, $\alpha_5$ and f-$CH_4$ for both lake basins were also similar to those reported in our previous study on Lake Tantaré Basin B (Clayer et al., 2018).

## 4 Discussion

### 4.1 Organic matter mineralization pathways at the sampling sites

The porewater data as well as the combined modeling of carbon isotopes and concentration profiles, allows to highlight key
OM mineralization mechanisms and to quantify the relative contribution of methanogenesis and fermentation to OM degradation at both sampling sites The $^{13}C$ isotopic signatures, i.e., highly negative values of $\delta^{13}C$-$CH_4$ and large differences between $\delta^{13}C$-$CO_2$ and $\delta^{13}C$-$CH_4$ (section 3.3 and Fig. S2 in the SI), as well as the modeling of the $\delta^{13}C$-$CO_2$ and $\delta^{13}C$-$CH_4$ profiles (section S2.2.2.1 and Fig S4a and b in the SI) all point to hydrogenotrophy as being the only pathway for methanogenesis in the two lake basins. The dominance of hydrogenotrophy is consistent also with the finding that acetate
concentrations were close to or below DL in the porewater samples. Under the condition that acetocalsty is negligible (i.e., $x = \nu_1$), reaction r1 from Table 1 becomes:

$$C_xH_yO_z + (2x - z)H_2O \xrightarrow{R_1} xCO_2 + (2x + \frac{y}{2} - z)H_2 \qquad (9)$$

Methanogenesis was also reported to be essentially hydrogenotrophic in the sediments of Basin B of Lake Tantaré (Clayer et al 2018). The absence of acetoclasty in the sediments of the oligotrophic lakes Bédard and Tantaré is consistent with the consensus that hydrogenotrophy becomes an increasingly important $CH_4$ production pathway: i) when labile OM is depleted
(Whiticar et al., 1986; Chasar et al., 2000; Hornibrook et al., 2000), ii) with increasing sediment/soil depth (Hornibrook et al., 1997; Conrad et al., 2009), or iii) with decreasing rates of primary production in aquatic environments (Wand et al., 2006; Galand et al., 2010).

The modelling of concentrations and $\delta^{13}C$ profiles revealed that oxidative processes occurred essentially in the upper 7 cm of the sediments of the perennially oxygenated Lake Tantaré Basin A, i.e., mainly in the $Z_1$ and, to a lesser extent, in the $Z_2$ (Table
3 and sections S2.1.2.1 and S2.1.2.2 of the SI). Moreover, it showed that methanotrophy was the dominant oxidative reaction in these sediment layers since 75% of the oxidants were consumed through r5 (section S2.2.2.2 of the SI). This outcome is consistent with several studies showing that methanotrophy occurs at higher rates than OM oxidation at low EA concentrations (Sivan et al., 2007; Pohlman et al., 2013; Kankaala et al., 2013; Thottahil et al., 2019). Methanotrophy is also evidenced in the $Z_1$ of this lake basin by the negative $R_{net}^{CH_4}$ value and by a shift of the $\delta^{13}C$-$CH_4$ profiles to more positive values in their upper
part (Fig. 2b and g). Use of Eq. 2 to model the EAs profiles with the code PROFILE predicts that $O_2$ was by far the main EA





involved either directly, or indirectly via the coupling with the Fe or S cycles, in the oxidative processes. Indeed, comparing the values of $R_{net}^{O2}$ and $R_{net}^{Ox}$ (see Section 3.2 and Table 2) shows that $O_2$ accounts for 87% and 70% of the oxidants consumed in the $Z_1$ and $Z_2$ of Lake Tantaré Basin A, respectively. Since $O_2$ penetration in the sediment by molecular diffusion is limited to ~4-mm, a significant amount of $O_2$ is predicted by Eq. 2 to be transported deeper in the sediment through bioirrigation. The

predominance of $O_2$ among the EAs consumed in the sediments is consistent with our previous study in this basin of Lake Tantaré (Clayer et al., 2016). Given that methanotrophy is the dominant oxidative process and that $O_2$ is the main oxidant consumed, it is probable that aerobic oxidation of methane prevails over its anaerobic counterpart in this lake basin. This is in line with the common thinking that $CH_4$ oxidation in freshwater lake sediments is carried out by methanotrophs essentially in the uppermost oxic sediment layer (Bastviken et al., 2008 and references therein).

The sharp upward depletion in $^{13}C$-$CH_4$ leading to a minimum $\delta^{13}C$-$CH_4$ value at 2.5 cm depth in Lake Tantaré Basin A sediments (Fig. 3a) was unanticipated since, according to the modeling with the code PROFILE, it occurs in the methanotrophic zone, i.e., where the remaining $CH_4$ is expected to be $^{13}C$-enriched as a result of $CH_4$ oxidation. Marked $^{13}C$-$CH_4$ depletions at the base of the sulfate-methane transition zone, where $CH_4$ is consumed via $SO_4^{2-}$ reduction, have often been observed in marine sediments (Burdige et al., 2016 and references therein). Such features are generally attributed to the production of $CH_4$

by hydrogenotrophy from the $^{13}C$-depleted DIC resulting from the anaerobic $CH_4$ oxidation, a process referred to as intertwined methanotrophy and hydrogenotrophy (e.g., Borowski et al., 1997; Pohlman et al., 2008; Burdige et al., 2016). Here the modelled $\delta^{13}C$-$CH_4$ profile captured the minimum in $\delta^{13}C$-$CH_4$ in the $Z_1$ by simply assuming concomitant hydrogenotrophy and methanotrophy in this zone and an upward-increasing $\alpha_4$ value from 1.085 in the $Z_3$ to 1.094 in the $Z_1$ (section S2.2.1 of the SI). These $\alpha_4$ values remain within the range reported for this isotope fractionation factor (Table S1 in the SI). A small

variation with sediment depth in the fractionation factor $\alpha_4$ is arguably possible since its value depends on the types of microorganisms producing $CH_4$ (Conrad 2005). The possibility that a depth variation in this isotope fractionation factor could explain some of the minima in $\delta^{13}C$-$CH_4$ reported in other studies should be considered.

In the $Z_2$ of Lake Bédard, the net rate of DIC production (i.e., 167 fmol cm$^{-3}$ s$^{-1}$) was more than 3 times that of $CH_4$ production (50 fmol cm$^{-3}$ s$^{-1}$; Table 2). Given that the $R_{net}^{Ox}$ was negligible in this zone (i.e., $R_5 = R_6 = 0$), we obtain from Eqs 3 and 4 and

Table 2 that $R_{net}^{CH_4} = R_4 = 50$ fmol cm$^{-3}$ s$^{-1}$ and $R_{net}^{DIC} = R_1 + R_2 - R_4 = 167$ fmol cm$^{-3}$ s$^{-1}$ (see section S2.1.2.2 of the SI). Should we assume that DIC production by r2 is negligible, i.e., $R_2 = 0$, a $R_1/R_4$ ratio of 4.3 would be obtained. This high ratio indicates that DIC was not produced by hydrogenotrophy (r4) coupled to fermentation (r1) alone in the $Z_2$ of this lake. Indeed, methanogenesis through the coupling of these two reactions yields a $R_1/R_4$ ratio of 2 if the fermenting substrate is carbohydrates (COS of 0) and lower than 2 if the fermenting substrate has a negative COS value. We thus attributed the

production of the additional DIC to the partial fermentation of HMW OM, an assumed non-fractionating process reported to occur in wetlands (Corbett et al., 2015). The better fitting of the $\delta^{13}C$-DIC profile when $\alpha_2$ is set to 0.980–0.984 rather than to 1.000 in the $Z_2$ (compare the blue and red lines in Fig. 4b) suggests that C fractionates during this partial fermentation process. Table 3 displays the depth-integrated reaction rates ($\Sigma R_i$) over the top 21cm of the sediment column which are given by:





$$\Sigma R_i = \sum_{j=1}^{3} \Delta x_j R_i \qquad (10)$$

where $\Delta x_j$ (cm) is the thickness of the zone $Z_j$. In this calculation, we assume that other zones of $CH_4$ or DIC production are

absent below 21 cm. Values of $\Sigma R_i$ clearly show that anaerobic carbon mineralization reactions (fermentation and methanogenesis) are important contributors to the overall OM mineralization in the two studied lake basins. Indeed, the sum of the rates of $CH_4$ production ($\Sigma R_4$), DIC production due to $CH_4$ formation ($\Sigma R_1 - \Sigma R_4$) and HMW OM partial fermentation ($\Sigma R_2$) represents 49% and 100% of the total OM degradation rate ($\Sigma R_1 + \Sigma R_2 + \Sigma R_5 + \Sigma R_6$) in the sediment of lakes Tantaré Basin A and Bédard, respectively. The contribution of anaerobic mineralization for Lake Tantaré Basin A is about 1.6 times

higher than the average of 30% reported for this lake basin in a previous study (Clayer et al., 2016). This significant discrepancy arises because these authors, in the absence of isotopic data to adequately constrain the $R_i$ values, assumed that $R_4 = 0$ in the net methanotrophic zone $Z_1$. Should we make the same assumption in the present study, we would also estimate that fermentation and methanogenesis represent only 30% of the total rate of OM degradation in the oxygenated Lake Tantaré Basin A and we would thus underestimate the importance of methanogenesis. The inclusion of $\delta^{13}C$ data in the present

modeling study thus allowed to better constrain the effective rates of $CH_4$ production ($R_4$).

### 4.2 Organic substrates for methanogenesis at the sampling sites

Table 3 indicates that hydrogenotrophy (r4) coupled to the complete fermentation of OM (r1) produces $CH_4$ at higher rates ($R_4$) than DIC ($R_1 - R_4$) in the $Z_1$ and $Z_2$ of both lake basins. This outcome is inconsistent with the equimolar production of $CH_4$ and DIC expected from the fermentation of glucose ($C_6H_{12}O_6$), the model molecule used to represent labile OM in

diagenetic models (Paraska et al., 2014), thus suggesting that the fermentation of this compound is not the exclusive source of the $H_2$ required for hydrogenotrophy. Had OM been represented by $C_6H_{12}O_6$ in r1, the rate of $H_2$ production by this reaction would have been twice that of $CO_2$, i.e., $2R_1$. For its part, the rate of $H_2$ consumption through hydrogenotrophy is four times that of the $CH_4$ production, i.e., $4R_4$. Hence, an additional $H_2$ production at rates of up to 212 and 70 fmol $cm^{-3}$ $s^{-1}$, i.e., $4R_4 - 2R_1$, is needed to balance the $H_2$ production rate expected from the fermentation of $C_6H_{12}O_6$ and the $H_2$ consumption rate by

hydrogenotrophy observed in the sediments of Lake Tantaré Basin A and Lake Bédard, respectively. As discussed by Clayer et al. (2018), this additional production rate of $H_2$ could be provided by a cryptic Fe-S cycle such as r8 (Table1), or by the production of $CH_4$ via the fermentation of organic substrates more reduced than glucose.

The progressive downward increases in dissolved Fe and $SO_4^{2-}$ (Fig. 2e, f, m and n) below ~5 cm depth and decrease in $\Sigma S(-II)$ (Fig. 2n) observed in the porewaters support a production of $H_2$ from r8 in both lakes. However, modeling the appropriate

solute profiles with the code PROFILE indicates that the production rates of dissolved Fe (<10 fmol $cm^{-3}$ $s^{-1}$) and $SO_4^{2-}$ (<1 fmol $cm^{-3}$ $s^{-1}$) and the consumption rate of $\Sigma S(-II)$ (<1 fmol $cm^{-3}$ $s^{-1}$) are about one order of magnitude too low to explain the missing $H_2$ production rate in both basins. Moreover, in the $Z_1$ and $Z_2$ of Lake Tantaré Basin A, the rate of solid Fe(III) reduction (<3 fmol $cm^{-3}$ $s^{-1}$; calculated from Liu et al. 2015) is much lower than that required from r8 (i.e., 1 to 2 times the





additional $H_2$ production of $4R_4 - 2R_1$; 70–424 fmol cm$^{-3}$ s$^{-1}$) to produce sufficient amounts of $H_2$ to sustain the additional

hydrogenotrophy. Given these results, we submit that a cryptic Fe-S cycle, if present, would contribute only minimally to the

missing rate of $H_2$ production, and that the fermentation of reduced organic compounds could provide a better explanation to

the imbalance between the $H_2$ production and consumption rates.

Since $CH_4$ is produced by hydrogenotrophy in the two lake basins ($\chi_H = 1$), Eqn. S15 (section S2.2.2. of the SI) describing

the COS of the fermenting organic substrate $C_xH_yO_z$ simplifies as:

$$COS = -4 \left( \frac{2 \left( R_{net}^{CH_4} - \frac{1}{2}\chi_M R_{net}^{Ox} \right) - R_1}{R_1} \right) \tag{11}$$

where $\chi_M$ is the fraction of oxidants consumed through methanotrophy. Combining Eqs. S7 and S5 of the SI with Eq. 11, we

obtain:

$$COS = -4 \left( \frac{R_{net}^{CH_4} - R_{net}^{DIC} - R_{net}^{Ox} + R_2}{R_{net}^{DIC} + R_{net}^{CH_4} + (1 - \chi_M)R_{net}^{Ox} - R_2} \right) \tag{12}$$

Introducing the values of $R_{net}^{CH_4}$, $R_{net}^{DIC}$, $R_{net}^{Ox}$ and $R_2$ (Table 2 and 3) into Eq. 12, we calculate COS values of −3.2 and −0.9 for

the $Z_1$ and $Z_2$ of Lake Tantaré Basin A, respectively, and of −1.0 to −1.1 for the $Z_1$ of Lake Bédard, respectively. Note that we

were unable to constrain with Eq. 12 the COS for the $Z_2$ of Lake Bédard since we had to assume a COS value to estimate $R_2$

and the COS has no influence of the modelled $\delta^{13}$C profiles (section S2.2.2.3 of the SI). Negative COS values between −0.9

and −1.1 suggest that fermenting OM in the sediments of the two lake basins would be better represented by a mixture of fatty

acids and fatty alcohols than by carbohydrates, as suggested by Clayer et al. (2018) for the sporadically anoxic Lake Tantaré

Basin B. For its part, the highly negative COS value of −3.2 calculated for the $Z_1$ of Lake Tantaré Basin A is unreasonable,

and the inaccuracy of the COS determination in this lake basin is discussed in section 4.3.

**4.3 Reduced organic compounds as methanogenic substrates in lake sediments**

In order to better appraise the COS of the fermenting OM in lakes, relevant datasets of porewater solute concentration profiles

were gathered from our data repository and from a thorough literature search. To be able to obtain by reactive-transport

modeling the $R_{net}^{solute}$ required to calculate the COS with Eq. 12, the datasets had to: (i) comprise porewater concentration

profiles of $CH_4$ and DIC and, ideally, those of the EAs; (ii) reveal a net methanogenesis zone, and iii) enable the carbonate

precipitation/dissolution contribution to the DIC concentrations to be estimated. Detailed information on the origin and

processing of the 17 selected datasets, acquired in 6 different lake basins from one sub-alpine and three boreal lakes sampled

at various dates and/or depths, is given in section S3 of the SI. The $CH_4$ and DIC porewater profiles determined at hypolimnetic

sites of these lake basins and their modeling with the code PROFILE are shown in Fig. 4, whereas the $R_{net}^{CH_4}$, $R_{net}^{DIC}$ and $R_{net}^{Ox}$

values determined from this modeling are regrouped in Table 4. The COS values displayed in Table 4 for all lake basins and

dates were calculated by substituting the appropriate $R_{net}^{CH_4}$, $R_{net}^{DIC}$ and $R_{net}^{Ox}$ values in Eq. 12 and assuming that $R_2 = 0$. This latter





assumption was not required Lake Tantaré Basin A (October 2015) and Lake Bédard (October 2015) for which $R_2$ values were known (Table 4). Equation 12 indicates that any DIC contribution from r2 would yield lower COS values than those reported in Table 4. The value of $\chi_M$ was assumed to be alternately 0 and 1 to provide a range of COS values. The only exception was Lake Tantaré Basin A in October 2015 for which $\chi_M$ is known to be 0.75 (section S2.2.2.2 of the SI). Note that although Eq.

12 was derived with the assumption that methanogenesis was hydrogenotrophic ($\chi_H = 1$), assuming that $CH_4$ was produced by acetoclasty ($\chi_H = 0$) would yield the same expression.

According to Table 4 the COS values are systematically negative at all dates for Lake Tantaré Basin B, Lake Bédard, Jacks Lake and the two sites of Lake Lugano, and they vary generally between −0.9 and −1.9, with the exception of a value of −2.5 obtained for Lake Tantaré Basin B in July 2007. This latter value is likely too low to be representative of fermenting material

and should be rejected. The mean (± SD) COS values are −1.7 ± 0.4 for Lake Tantaré Basin B, −1.4 ± 0.4 for Lake Bédard, −1.4 ± 0.2 for Jacks Lake and −1.4 ± 0.3 for Lake Lugano. These COS values, representative of a mixture of fatty acids (COS of −1.0 for C4-fatty acids to about −1.87 for C32-fatty acids) and of fatty alcohols (COS = −2.00), strongly supports the idea that methanogenesis in boreal lakes sediments, as well as in the sediments of other types of lakes, is fueled by more reduced organic compounds than glucose. Lipids such as fatty acids and fatty alcohols with similar COS are naturally abundant in

sediments to sustain the estimated rates of $CH_4$ and DIC production during fermentation (Hedges and Oades, 1997; Cranwell, 1981; Matsumoto, 1989; Burdige, 2006). As discussed by Clayer et al. (2018) the most labile organic compounds (i.e., proteins and carbohydrates) can be rapidly degraded during their transport through the water column and in the uppermost sediment layer, leaving mainly lipids as metabolizable substrates at depths where fermentation and methanogenesis occurs. This interpretation is consistent with thermodynamic and kinetic evidences that proteins and carbohydrates are more labile and are

degraded faster than lipids (LaRowe and Van Cappellen, 2011).

The COS values determined for the perennially oxygenated Basin A of Lake Tantaré (mean of −0.6 ±1.1; range of −3.2 to 2.1; Table 4) are much more variable than for the five other lake basins which undergo seasonal anoxia. Moreover, the COS values estimated for October 2015 in the $Z_1$ (−3.2), September 2016 (0.4–0.6) and October 2005 (1.8–2.1) are unrealistic. Indeed, the very negative value of −3.2 does not correspond to any degradable compound under anoxic conditions, whereas the positive

values of 0.4–0.6 and 1.8–2.1 would involve either amino acids and nucleotides which are very labile (Larowe and Van Cappellen 2011) and tend to be degraded in the water column (Burdige 2007) or oxidized compounds, such as ketones, aldehydes and esters, known to be quickly reduced to alcohols. These observations indicate that the COS determination in this lake basin is unreliable. The misestimation of the COS can probably not be explained by the presence of $O_2$ itself at the sediment surface of Lake Tantaré Basin A. Indeed, the sediment surface was also oxic at the sites Melide and Figino of Lake

Lugano in March 1989 (Table 4) as revealed by detectable bottom water $[O_2]$ (Table 4), and by low [Fe], undetectable $\Sigma S(-II)$ and $[CH_4]$ and relatively high $[SO_4^{2-}]$ in overlying water (Lazzaretti et al., 1992; Lazzaretti-Ulmer and Hanselmann, 1999). Despite this, the COS values determined for the two sites of Lake Lugano appear to be realistic and coherent with those calculated for Lakes Tantaré Basin B, Bédard and Jacks. However, we know that benthic organisms are present in Lake Tantaré Basin A (Hare et al., 1994) but lacking at the two sites of Lake Lugano, as shown by the presence of varves (Span et al., 1992)





and the absence of benthos remains in the recent sediments at these sites (Niessen et al., 1992). Clayer et al. (2016) provided
evidences that sediment irrigation by benthic animals is effective in Lake Tantaré Basin A and that it should be taken into
account in modeling the porewater solutes profiles. However, these authors also point out the difficulty to properly estimate
the magnitude of solute transport by bioirrigation. The term used in Eq. 2 to calculate this contribution, i.e., $\varphi\alpha_{irrigation}$
($[solute]_{tube} - [solute]$), is indeed an approximation of intricate 3-D processes (Meile et al. 2005). And, in the conceptualization
of this bioirrigation term, it was notably assumed that benthic animals continuously irrigate their tubes to maintain solute
concentrations in their biogenic structures ($[solute]_{tube}$) identical to those in the water overlying the sediments. But
microbenthic animals are generally reported to irrigate the sediments in a discontinuous manner and the solute concentrations
in their biogenic structures may be highly variable with time (Boudreau and Marinelli 1994; Forster and Graf 1995; Riisgård
and Larsen 2005; Gallon et al. 2008). Hence, owing to the imperfection of the representation of bioirrigation in Eq. 2, COS
values estimated for the sediment of Lake Tantaré Basin A should be treated with caution, especially in the $Z_1$ where the
bioirrigation coefficient takes the highest value. Another potential bias in the estimation of COS values for the oxygenated
basin is the possibility of DIC production through HMW OM fermentation (reaction r2; Corbett et al. 2013). Note that fitting
with Eq. 6 the experimental $\delta^{13}$C data does not allow partitioning the production of DIC between r1 and r2 since the two
processes share the same value of fractionation factor ($\alpha_1 = \alpha_1 = 1.000$). It was possible to attribute unequivocally the excess
of DIC production rate over that of $CH_4$ production in the $Z_2$ of Lake Bédard in October 2015 (Table 4 and Section S2.1.2.2
of the SI) to HMW OM fermentation merely because $R_{net}^{Ox}$ was negligible compared to $R_{net}^{CH_4}$ and  $R_{net}^{DIC}$ , which is not the case
for Lake Tantaré Basin A (Table 4). Equation 12 indicates that to obtain negative COS values for Lake Tantaré Basin A in
September 2006 and October 2005, $R_2$ should be >11 fmol cm$^{-2}$ s$^{-1}$ and >110 fmol cm$^{-2}$ s$^{-1}$, respectively. These $R_2$ values
correspond to transferring >9% and >44% of the rate of DIC production from $R_1$ to $R_2$ for September 2006 and October 2005,
respectively. The above discussion underlines several factors that can explain the unreliability in the actual COS estimation
for the perennially oxic Lake Tantaré Basin A, and further research is needed to better assess the importance of these factors.
However, it does not dismiss that the substrate for methanogenesis in this lake basin may have a negative COS value.

**5 Conclusions**

Our results show that fermentation and methanogenesis represent nearly 50% and 100% of OM mineralization in the top 25
cm of the sediments at the hypolimnetic sites in Lake Bédard and in Basin A of Lake Tantaré, respectively and that methane
is produced only by hydrogenotrophy at these two sites. An earlier study reached similar conclusions about the pathways of
methanogenesis and the contribution of this process in OM mineralization in Basin B of Lake Tantaré (Clayer et al. 2018).
Reactive-transport modelling of porewater solutes from three boreal lakes, i.e., Bédard, Tantaré (Basin B) and Jacks, as well
as of the sub-alpine Lake Lugano (Melide and Figino sites) consistently showed that the main substrates for sediment
methanogenesis at deep seasonally anoxic hypolimnetic sites have a mean COS value of $-1.4 \pm 0.3$. Mineralization of the most





labile compounds during OM downward migration in the water column and in the uppermost sediment layers likely explains why reduced organic compounds fuel methanogenesis in these sediments.

The current representation of the fermenting OM, i.e., $CH_2O$, in process-based biogeochemical models entails a significant risk of misestimating sedimentary $CH_4$ and $CO_2$ production and release to the bottom water and, to a certain extent, of

mispredicting evasion of these greenhouse gases to the atmosphere under transient environmental scenarios. To better constrain $CH_4$ and $CO_2$ production within sediments, we suggest taking specifically into account the COS of the fermenting OM in formulating the reactions of methanogenesis associated with fermentation in these models. For example, the rates of $CH_4$ ($R^{CH_4}$) and DIC ($R^{DIC}$) production during fermentation coupled to hydrogenotrophy can be expressed as:

$$R^{CH_4} = R_4 = \frac{4 - COS}{8} R_1 \qquad (13)$$

$$R^{DIC} = R_1 - R_4 = R_1 \left(1 - \frac{4 - COS}{8}\right) \qquad (14)$$

Given these rate expressions, the stoichiometric formulation of a typical fermentation reaction producing methane becomes:

$$CH_a O_b \rightarrow \frac{4 - COS}{8} CH_4 + \left(\frac{4 + COS}{8}\right) CO_2 \qquad (15)$$

where $a = 2 - \frac{COS}{2}$, $b = 1 + \frac{COS}{4}$. Note that the same stoichiometric formulation would be obtained for acetoclastic methanogenesis.

The approach used to estimate the COS of the fermenting OM, although successful for the seasonally anoxic basins, failed to produce reliable COS values when applied to the perennially oxygenated Basin A of Lake Tantaré. We attribute this peculiarity to a misestimation and/or misrepresentation of the benthic irrigation and to the impossibility to partition the DIC production

between reactions r1 and r2 which share the same fractionation factor value. Similar problems would likely be encountered also in other lake ecosystems such as epilimnetic sediments and wetlands where solute transport processes remain ill-known. Indeed, these shallow aquatic environments are subject to enhanced benthic activity (Hare 1995), to plant-mediated transport of $CH_4$ and $O_2$ (Chanton et al. 1989; Wang et al. 2006), as well as to turbulence (Poindexter et al. 2016) which complicates the estimation of $CH_4$ and $CO_2$ production and consumption rates. Hence, the remaining challenge resides in the robust estimations

the COS of the fermenting OM in epilimnetic sediments and shallow freshwater environments (e.g., ponds, wetlands), since these environments were shown to be the main contributors to freshwater $CH_4$ release to the atmosphere (Del Sontro et al., 2016, Bastviken et al., 2008). One potential solution is to investigate trends in the oxygen isotope signatures in the sedimentary DIC in addition to $\delta^{13}C$ values since it is also influenced by the source of the OM undergoing degradation (e.g., Sauer et al., 2001).


## Data availability:

Upon acceptance, readers will be able to access the data at this url: https://www.hydroshare.org/resource/38e069761d7b4cf4abe3cbcaaac06016/. A proper reference with a DOI will be made available to cite this dataset if the present paper is accepted.

## Author contribution:

Conceptualization: FC, AT, and CG. Data curation: FC and AT. Formal analysis: FC and AT. Funding acquisition: CG, YG and AT. Investigation: FC and YG. Methodology: AT, CG, YG and FC. Project administration: CG. Resources: CG and YG. Software: FC. Supervision: CG, AT and YG. Validation: AT. Writing – original draft: FC and AT. Writing – review & editing: All.

## Competing interests:

The authors declare that they have no conflict of interest.

## Acknowledgments

We thank P. Boissinot, L. Rancourt, P. Girard, J.-F. Dutil, S. Duval, A. Royer-Lavallée, A. Laberge and A. Barber for research and field work assistance. We are also thankful to J.-F. Hélie, from the Laboratoire de géochimie des isotopes stables légers (UQÀM), who graciously calibrated our $\delta^{13}$C internal standard. This work was supported by grants to C.G., A.T. and Y.G. from the Natural Sciences and Engineering Research Council of Canada and the Fonds de Recherche Québécois – Nature et Technologies. Permission from the Québec Ministère du Développement durable, de l'Environnement et de la Lutte contre les changements climatiques to work in the Tantaré Ecological Reserve is gratefully acknowledged.

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





Figure 1: Location map and bathymetry of Lakes Tantaré and Bédard. The bathymetric map of Lake Tantaré was reproduced from
the map C-9287 of the Service des eaux de surface of the Québec Ministry of Environment. The map of Lake Bédard was reproduced
from D'Arcy (1993). Dioxygen concentrations in the water column of Lake Tantaré basins A and B, and of Lake Bédard are given
for June (black lines) and October (red lines).

**Figure 2 : Replicate porewater profiles of CH₄ (a and i), δ¹³C-CH₄ (b and j), DIC (c and k), δ¹³C-DIC (d and l), SO₄²⁻ (e and m), Fe and ΣS(−II) (f and n), and comparison of the modeled (blue lines) and average (n = 3) measured (symbols) concentration profiles of CH₄ (g and o) and DIC (h and p) in Lakes Tantaré Basin A (a–h) and Bédard (i–p). Different symbols indicate data from different peepers and empty symbols are for concentrations below detection limit. The horizontal dotted line indicates the sediment-water interface. The thick and thin blue lines represent the net solute reaction rate (R$_{net}^{solute}$) and the modeled concentration profiles, respectively. The red area fills correspond to the sediment zones Z₂.**




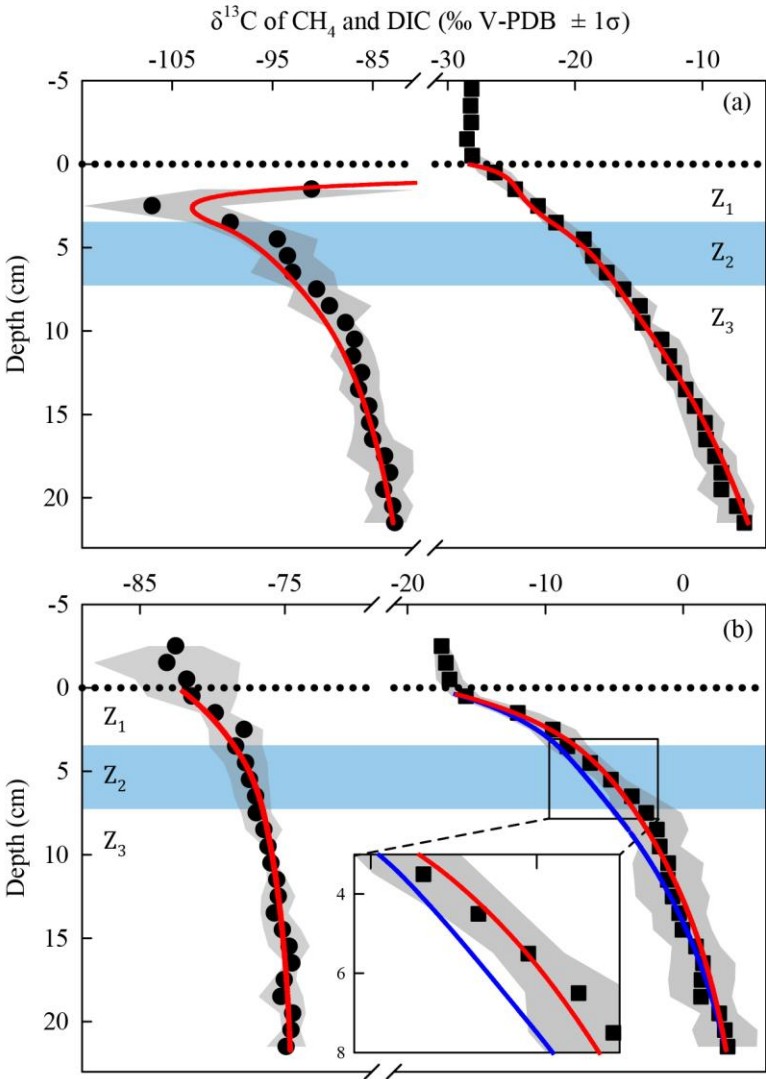

**Figure 3 : Comparison of the simulated (lines) and measured average (n = 3) δ¹³C profiles of CH₄ (circles) and DIC (squares) in the**
**porewater of Lake Tantaré Basin A (a) and Lake Bédard (b). The horizontal dotted line indicates the sediment-water interface. The**
**variability in δ¹³C values (± one standard deviation – σ) related to the spatial heterogeneity within the sampling area is shown by the**
**grey area fills. The zone Z₂ is delimited by the blue area fill. In panel b, the blue lines are the profiles simulated with the default rate**
**values and optimal αᵢ and f values as described in section S2.2.1. The red lines in panel (b) are the profiles simulated with α₂ values**
**of 0.980–0.984 (see section 4.1 for details).**



**Figure 4 :** Comparison of the modeled (blue lines) and average (n = 3) measured concentration profiles of CH₄ (squares) and DIC (circles) in Lakes Tantaré Basin A (a–d) and Basin B (a–h), Bédard (i), Jacks Lake (j–k) and Lake Lugano (l–o) at various sampling dates. The thick red lines represent the net solute reaction rate ($R_{net}^{solute}$).



**Table 1: Reactions (r1–r8) considered, their reaction rates ($R_1$–$R_8$) and carbon isotopic fractionation factors ($\alpha_1$–$\alpha_7$).**

| Description | Reaction | ID |
|---|---|---|
| **$CO_2$ production due to complete fermentation of labile OM [a]** | $C_xH_yO_z + (x + \nu_1 - z)H_2O \xrightarrow[\alpha_1]{R_1} \left(\frac{x - \nu_1}{2}\right)CH_3COOH + \nu_1 CO_2 + \left(\frac{y}{2} - z + 2\nu_1\right)H_2$ | r1 |
| **$CO_2$ production due to partial fermentation of HMW OM [a,b]** | $\nu_2 HMW\ OM \xrightarrow[\alpha_2]{R_2} \nu_3\ LMW\ OM + \nu_4 CO_2$ | r2 |
| **Methanogenesis via** | | |
| acetoclasty | $CH_3COOH \xrightarrow[\alpha_3]{R_3} CH_4 + CO_2$ | r3 |
| hydrogenotrophy | $CO_2 + 4H_2 \xrightarrow[\alpha_4]{R_4} CH_4 + 2H_2O$ | r4 |
| **$CO_2$ production due to** | | |
| methanotrophy | $CH_4 + 2\ Oxidants \xrightarrow[\alpha_5]{R_5} CO_2 + 2\ Reducers$ | r5 |
| OM oxidation | $OM + Oxidant \xrightarrow[\alpha_6]{R_6} CO_2 + Reducer$ | r6 |
| **Precipitation of siderite** | $Fe^{2+} + CO_2 + H_2O \xrightarrow[\alpha_7]{R_7} FeCO_{3(s)} + 2H^+$ | r7 |
| **$H_2$ production through a Fe-S cryptic cycle [a,c]** | $(16 + \nu_5)H_2S + 8FeOOH \xrightarrow{R_8} 8FeS_2 + \nu_5 SO_4^{2-} + (4 + 4\nu_5)H_2 + (16 - 4\nu_5)H_2O + 2\nu_5 H^+$ | r8 |

[a] where $\nu_1$ can have any value between 0 and x, values for $\nu_2$–$\nu_4$ are unknown and $\nu_5$ can have any value between 0 and 1.

[b] HMW OM and LMW OM designate high and lower molecular weight organic matter, respectively.

[c] adapted from Holmkvist et al. (2011)





**Table 2: Net production rates ( $R_{net}^{solute}$ ) of CH$_4$, DIC and oxidants obtained with the code PROFILE in the three CH$_4$**
**consumption/production zones (Z$_1$, Z$_2$ and Z$_3$) for both sampling sites.**

| Sampling site ([O$_2$] in mg L$^{-1}$) | Zones | Depth (cm) | $R_{net}^{DIC}$ | $R_{net}^{CH_4}$ (fmol cm$^{-3}$ s$^{-1}$) | $R_{net}^{Ox}$ |
|---|---|---|---|---|---|
| Tantaré Basin A (2.5) | Z$_1$ | 0–3.6 | 223 | −7 | −335 |
| | Z$_2$ | 3.6–7.2 | 113 | 39 | −103 |
| | Z$_3$ | 7.2–21.5 | −2 | 1 | |
| Bédard (<0.1) | Z$_1$ | 0–3.6 | 65 | 100 | −6.5 |
| | Z$_2$ | 3.6–7.2 | 167 | 50 | −4.5 |
| | Z$_3$ | 7.2–21.5 | −13 | 5 | |





**Table 3: Molecular diffusivity ratio of CH₄ (f-CH₄) as well as the isotopic fractionation factors ($\alpha_1$, $\alpha_2$, $\alpha_4$–$\alpha_7$) and rates (R₁, R₂, R₄– R₇; fmol cm$^{-3}$ s$^{-1}$) of each reaction involved in OM mineralization in each zone and for the whole sediment column (ΣR$_i$ ; fmol cm$^{-2}$ s$^{-1}$) corresponding to the lowest values of N$_{res}$. At both study sites, R₃ was shown to be negligible. See section S2 of the SI for details.**

| Study site | Zones | f-CH₄ | $\alpha_1$ | $\alpha_2$ | $\alpha_4$ | $\alpha_5$ | $\alpha_6$ | $\alpha_7$ | R₁ | R₂ | R₄ | R₅ | R₆ | R₇ |
|---|---|---|---|---|---|---|---|---|---|---|---|---|---|---|
| Tantaré Basin A | Z₁ | 1.003 | 1.000 | - | 1.094 | 1.024 | 1.000 | - | 132 | - | 119 | 126 | 84 | - |
| | Z₂ | 1.003 | 1.000 | - | 1.087 | 1.005 | 1.000 | - | 126 | - | 78 | 39 | 26 | - |
| | Z₃ | 1.003 | - | - | 1.085 | - | - | - | - | - | 1 | - | - | - |
| | ΣR$_i$ | | | | | | | | 931 | - | 721 | 592 | 394 | - |
| Bédard | Z₁ | 1.003 | 1.000 | - | 1.074 | - | - | - | 165 | - | 100 | - | - | - |
| | Z₂ | 1.003 | - | 0.984[a] | 1.074 | - | - | - | 72[b] | 145[b] | 50 | - | - | - |
| | Z₃ | 1.003 | - | - | 1.074 | - | - | 0.995 | - | - | 5 | - | - | 8 |
| | ΣR$_i$ | | | | | | | | 853 | 522 | 612 | - | - | 114 |

[a]the optimal value of $\alpha_2$, given here is for a COS value of −1.5, varies slightly with the COS value (see section S2.2.2.3 of the SI).

[b]the value of R₁ and R₂, given here is for a COS value of −1.5, varies with the COS value (see section S2.2.2.3 of the SI).

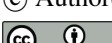



**Table 4: Net reaction rates ($R_{net}^{solute}$; fmol cm$^{-3}$ s$^{-1}$) of CH$_4$, DIC and oxidants in the zone with the highest production rate of CH$_4$ as well as the O$_2$ concentration in the bottom water ([O$_2$] in mg L$^{-1}$), the R$_2$ rates (fmol cm$^{-3}$ s$^{-1}$) and the average carbon oxidation state (COS) of the fermenting OM at the origin of CH$_4$ calculated with Eq. (12) at both study sites, Lake Tantaré Basin B (Fig. 1), Jacks Lake (Carignan and Lean 1991) and Lake Lugano (Lazzaretti-Ulmer & Hanselmann 1999) for various sampling dates.**

| Lake Basin | Sampling date | [O$_2$] | $R_{net}^{DIC}$ | $R_{net}^{CH_4}$ | $R_{net}^{Ox}$ | R$_2$ | Reference | COS[a] | |
|---|---|---|---|---|---|---|---|---|---|
| | | | | | | | | Min. | Max. |
| Tantaré Basin A, 15 m | Oct 2015 – Z$_1$ | 3.5 | 223 | −7 | −335 | 0 | this study | −3.2 | −3.2 |
| | Oct 2015 – Z$_2$ | 3.5 | 113 | 39 | −103 | 0 | this study | −0.9 | −0.9 |
| | Jul 2012 | 6.0 | 143 | 245 | −66 | - | 1 | −2.1 | −1.7 |
| | Sep 2006 | 4.0 | 89 | 33 | −45 | - | 1 | 0.4 | 0.6 |
| | Oct 2005 | 3.1 | 202 | 48 | −44 | - | 1 | 1.8 | 2.1 |
| | Sep 2004 | 4.6 | 99 | 45 | −60 | - | 1 | −0.3 | −0.2 |
| Tantaré Basin B, 22 m | Oct 2014 | < 0.1 | 42 | 116 | −1 | - | 2 | −1.9 | −1.9 |
| | Oct 2011 | 0.4 | 279 | 783 | −12 | - | 1 | −2.0 | −1.9 |
| | Jul 2007 | 4.1 | 283 | 1147 | −20 | - | 1 | −2.5 | −2.5 |
| | Oct 2006 | < 0.1 | 442 | 825 | −2 | - | 1 | −1.2 | −1.2 |
| Bédard, 10 m | Oct 2015 – Z$_1$ | < 0.1 | 65 | 100 | −6.5 | 0 | this study | −1.1 | −1.0 |
| | Oct 2003 | < 0.1 | 205 | 408 | −13 | - | 3 | −1.4 | −1.4 |
| Jacks Lake, 15 m | Sep 1981 | na | 284 | 514 | - | - | 4 | −1.2 | −1.2 |
| Jacks Lake, 22 m | Sep 1981 | na | 904 | 2030 | - | - | 4 | −1.5 | −1.5 |
| Lugano, Melide, 85 m | Mar 1989 | 2.0 | 228 | 388 | −83 | - | 5 | −1.8 | −1.6 |
| Lugano, Melide, 85 m | Jun 1989 | < 0.1 | 45 | 97 | −1 | - | 5 | −1.5 | −1.5 |
| Lugano, Figino, 95 m | Mar 1989 | 4.0 | 1168 | 1903 | −234 | - | 5 | −1.4 | −1.3 |
| Lugano, Figino, 95 m | Jun 1989 | < 0.1 | 237 | 355 | −19 | - | 5 | −1.0 | −0.9 |

[a] Minimum and Maximum COS values were obtained by setting $\chi_M$ to 0 and 1 in Eq. (12), except for Tantaré Basin A in October 2015 for which $\chi_M$ is known to be 0.75.

References: (1) Clayer et al. (2016), (2) Clayer et al. (2018), (3) see Supporting Information, (4) Carignan and Lean (1991), (5) Lazzaretti-Ulmer & Hanselmann (1999).