# Peer review of "Mineralization of organic matter in boreal lake sediments: Rates, pathways and nature of the fermenting substrates"

_Biogeosciences, 2020_

## Referee Comment (RC1) · Anonymous Referee #1 · 23 Mar 2020

The manuscript is a modification of previous work published by Clayer et al 2018 in Geochimica et Cosmochimica Acta. It builds on the hypothesis that the degradation of organic material under anaerobic conditions has two ultimate sinks – CO2 or CH4, the most reduced and oxidized states of carbon. The relative abundance of these two should therefore provide information on the average oxidation state of the degraded organic material accounting for transport and other sources of CO2. This principal approach has been presented in Clayer et al (2018) GCA. The current paper is very similar to this published work and contains many data that are shared. It was not obvious to me where this work is a significant novel contribution beyond what has also ready been published. The authors use a simple steady state reaction transport model

for diffusive and advective transport to determine microbial process rates based on sediment porewater concentration gradients. Furthermore, the isotope composition of DIC is used to improve the mass balance calculations. Below I question the validity of this approach to obtain a meaningful mass balance using the Berg et al model at steady state.

This manuscript is difficult to understand manuscript and very technical in its description. What makes this manuscript so hard to read and understand is the multitude of R subscripts that are used in the text and the extensive treatment of the methodology in the appendix. The fractional equations are nowhere introduced. Deserves an explanation. In practice, one has to have a table on the side to look up what reaction a particular R subscript refers to and, in addition, know all the notations from Clayer et al (2018) to follow this work. Still, to follow the conclusions becomes increasingly confusing as one reads along, until one is either lost or exhausted. In the current form, the manuscript cannot be digested.

I recommend that the authors outline the hypothesis, mathematically, in the materials and methods section, of how their methodology allows them to get at the oxidation state of oxidized organic matter. In the current form, the reader has to wade through too much text to get to this most interesting point of the manuscript. This paper requires a much better didactic approach to get methods and goals across and the authors get sidetracked in many details that make it hard to follow their ultimate goal. It is, in its current form, not streamlined enough and requires very significant rewriting and restructuring to make the approach more understandable and possible to evaluate critically. At present, I cannot evaluate the quality of the manuscript, but am left in doubt about its novelty given the similarity to the 2018 GCA paper.

While the fundamental goal, to arrive at the oxidation state of metabolizable organic material, is of some significance, the presentation of the approach is not well developed and can be improved considerably.

Data and basic approach (although I did get lost in the complicated d13C treatment) are, in principle feasible, but overall I am concerned that the instrumental and modelling analytical uncertainty is too great to pin down the COS sufficiently (although an error is given). The authors provide statistical data to support their assertion, but it was not possible for me, based on the complicated description, to relate the outcome of these tests to the goal of the manuscript, i.e., the original oxidation of the degrading organic material. The authors must make sure, in a succinct and understandable and not too wordy fashion, how their methodology allows them to pin this value down sufficiently. Remove as much as possible reiterations of what has already been said and discussed in detail in Clayer et al 2018 GCA and restrict this paper to the novel information.

A lot of the discussion about the CH4 isotopes are not really part of the goal of this paper. This should be separated.

There are a few assumptions whose impact I don't understand or that are difficult to assess, e.g., that there are no anaerobic reoxidation reactions for sulfide; elemental sulfur with FeOOH. The paper does not lost O2 uptake rates for the oxic part of the year, which is an important constraint on the 'background CO2 levels in the buried pore-waters. The paper does not constrain oxygen penetration depths or the importance of bioturbation processes for DIC levels, and does not show O2 microelectrode profiles, which would be necessary to constrain the inorganic oxidative processes. Therefore the constraints for the diagenetic system, e.g., by having total O2 uptake rates are far and few. In principle, non-steady state reaction transport modelling with a much more advanced model are necessary to tackle this question, if it is possible at all.

Another curious observation is the omission of NO3 dynamics as part of O2 consumption by nitrification; also here there are no constraints on the system concerning NH4+ to accompany C mineralization dynamics.

The model also ignores O2 consumption due to Fe oxidation, but curiously the authors choose to include instead sulfide oxidation with O2 and FeOOH.

[Figure]

It seems very hard to see how the boundary conditions can be reasonably constrained to continue with the approach used by the authors.

Line comments: Line 139: There are more products than acetate $CO_2$ and $CH_4$ and $H_2$: Formate, propionate, isopropionate, lactate, butyrate, isobutyrate, pyruvate, succinate, etc.; The sum of the latter can be as high as 30% of the total VFA. Ok, acetate is low, but why should it not, if it is consumed by terminal oxidizers? L.155 Profile underestimates the oxidation rate because it is a fit of a net rate, not a gross rate, e.g., cryptic cycling leads to $CO_2$ production by sulfate reduction in the absence of a curvature in the gradient. Line 191: No good explanation for the low acetate concentrations? Line 281 Equation 9 has not been introduced previously. I don't get those fractions. Line 322: This sentence is confusing, why should hydrogenotrophy produce DIC coupled to fermentation? Only fermentation r1 may produce $CO_2$. L.330: To avoid confusion, a carbon mineralization process leads to the formation of a mineral acid, e.g., carbonic acid. Neither fermentation nor methanogenesis can therefore be called mineralization processes. They are carbon degradation/decomposition processes. Line 332: Please change your terminology Methanogenesis by hydrogenotrophy cannot lead to $CO_2$ formation Line 337-340 This conclusion cannot truly be validated with the approach used here. Line 353-355 Again, the authors make the mistake of modelling net concentration profiles to extract information on gross rates. The $H_2$ production and $CO_2$ production rates by cryptic cycling are not reflected in curvatures of concentration gradients, these only represented the net effect.

A cryptic sulfur cycle is only used to argue for $H_2$ production. Why not $CO_2$ production by sulfate reduction?

The overall problem with the approach is that a balance based on $CO_2$ and $CH_4$ and the OM oxidation state is too poorly constrained. In reality, in addition to a mass balance an independent charge balance should be achieved to constrain the original oxidation state of the OM. The current approach balances the electrons between the mass of methane and total $CO_2$ accounting for diffusive transport. This could likewise

be achieved by adjusting the alkalinity.

I think that the authors use the model the wrong way. It is perfectly fine for comparing rates in the different zones, but it is not possible to balance the inventories in the respective zones with this model. A more sophisticated reaction transport model that accounts for the cumulative amount of DIC formed during burial needs to be used to explain the amount and isotope composition of DIC. The model Profile only captures a snapshot of a concentration distribution, i.e., the steady state, and it does not allow for calculating cumulative effects during burial, which is important for a diagenetic model and for this case to account for the buried amount of DIC from oxic respiration. In addition, the steady state assumption is invalid for most natural cases except for very small distance, e.g., at the micrometer scale where diffusion is extremely fast. A time-dependent model that includes mass accumulation rates must be used here.

---

## Referee Comment (RC2) · Anonymous Referee #2 · 15 Apr 2020

The paper addresses an interesting and fundamentally important question: which fraction of sedimentary organic matter is mineralized through methanogenesis. Based on modeling and analyses of data from two lakes, it argues that organic carbon in negative oxidation states is used preferentially and the hydrogenotrophic pathway of methanogenesis dominates. If true, this may have profound implications for modeling the carbon cycle and interpretations of sedimentary signatures of carbon isotopes. Both the dataset and the model go well beyond the level of detail of typical diagenetic studies, which is indeed a requirement for figuring out the important fine details of organic matter mineralization.

[Figure]

This important work, however, could be improved in several key areas.

Style and clarity: The clarity of the narrative deteriorates towards the end of the manuscript. In particular, stating clearly and emphasizing throughout the text the main finding of the work would greatly improve readability. Inferences from modeling of the isotopic profiles could also benefit from a clearer presentation. Key statement such as (Line 265) "practically all CH4 is produced through hydrogenotrophy" are inferred from modeling d13C profiles, but I admit I was rather lost following the description, particularly trying to separate the relative contributions of hydrogenotrophic vs acetoclastic methanogenesis.

Originality: Much of the work is an update on the results of Clayer et al. 2018. The text should clearly distinguish the novel aspects, especially how (or if) the difference in conclusions is more than just refinement of the numbers from that previous work. For example, a statement on lines 58-60 reads: "Based on the observation that methanogenesis produced CH4 three times faster than CO2 . . .. Clayer et al. (2018) concluded that the fermenting OM had a markedly negative COS value of -1.9". This parallels the statement in the Abstract, which presumably should highlight the results from this work: "we calculate, from CH4 and DIC production rates. . .COS below -0.9". This seems to convey the same information.

Justifying the inclusion or omission of processes: The coupling with the sulfur cycle seems particularly suspect. The cryptic oxidation of sulfide coupled to iron oxides is used as an important pathway for H2 production. While this reaction is commonly considered (but can be written in various stoichiometries), it is rarely the only reaction that is considered from the complicated network of reactions that comprise the sedimentary Fe and S cycling. Puzzlingly, the modeled SO4 and Fe profiles are not shown (line235). These absolutely need to be shown. The sulfur cycle in this system seems highly unusual. For example (Line 201 and Fig. 2), "SO42- concentrations reach a minimum between SWI and 5 cm depth, and increase below". These highly unusual features need to be discussed. How can SO4 be produced in anoxic sediment? Does oxida-
tion of H2S by Fe(III) somehow proceed faster than sulfate reduction? What about precipitation of iron sulfides? Similarly, precipitation of CaCO3 does not seem to be considered as a CO2 sink, while Line 380 mentions that it had to be considered by the used datasets. Were the saturation indexes negative for the study sites?

Discussing implications: If the organic matter used in methanogenesis had negative COS, what happened to the rest of the C pool? Is oxidized OM not mineralized? Or is it mineralized preferentially earlier, in the water column? What are the implications, e.g. for burial, signature of OM in rock record, etc.? The statement on line 450 seems to address it somewhat, but the statement is not clear. It would also help to discuss how special or typical these lakes are, given that the implications seem to include global extrapolations. For example, diagenesis in Lake Tantare (or is it Lake Bedard? – see below) seems to lack contributions from terminal electron acceptors. How different would this be from a "typical" boreal forest lake?

Other criticisms and suggestions:

Conclusions: "fermentation and methanogenesis represent. . .100% of OM mineralization . . . in Lake Tantare" – Methanogenesis can be fermentation. More importantly, why are there no contributions from terminal electron acceptors? Is it really 100%? Confusingly, Fig. 2 shows that sulfate reduction is clearly active in Lake Tantare, whereas contributions of terminal electron acceptors are likely smaller in Lake Bedard.

One of the main results seems to be expressed by Eq. 15. Given the range of COS values (-1.4+-0.3), it might be helpful to state the range in the stoichiometric coefficients explicitly.

Line 284: "i) when labile OM is depleted, ii) with increasing sediment depth" – aren't these two statements in practice the same?

Line 454: "misestimating CH4 and CO4 production" – not sure what this means. Underestimating the amounts? But early diagenetic models generally work okay and can

reproduce measured profiles. Are the differences small enough that they are within uncertainties?

---

## Referee Comment (RC3) · Anonymous Referee #3 · 22 Apr 2020

In this paper, Clayer et al. found the average carbon oxidation state (COS) is negative COS values by modelling solute pore-water profiles. They concluded that carbohydrates do not adequately represent the fermenting OM and that the COS should be included in the formulation of OM fermentation in models. It is an interesting work and the results can guide new biogeochemical model for OM degradation.

However, the manuscript needs substantial improvement of the presentation before it can be recommended for publication. The main issues is the lack of the OM and mobile labile information. There are no data for the deposition/sedimentation rate of OM, the chemical composition of OM (C,H,O,N,S,P,..), d13C distribution of OM et al.

[Figure]

The results of COS from modelling solute pore-water profiles have not been validated. Even in the solute model there are too many fitting parameters and the conclusion is not convincing.

Here are some details:

1. Reactions: Since the reactions the precipitation of siderite (r7) and sulfide oxidation by iron oxides (r8) were taken into account, the pyrite formation by Fe2+ and H2S should be considered,too. The hydrogen H2 in eq.(r8) is usually consumed easily by sulphate reducer bacteria rather than CO2 reduction. The authors used general oxidant instead of O2,Fe(III) and SO4, which could have different oxidation rates, especially for CH4 oxidation (r5).

2. Rate calculation: The reaction rate were calculated by computer code PROFILE. This rates obtained from PROFILE were very rough. It is better to use reaction-transport model to calculate the rate by considering OM deposition and degradation.

3. The bioirrigation term was shown in the equation (2) but the bioirrigation depth and coefficient were not clear. How does the bioirrigation affect COS estimatation was also not clear. General once bioirrigation is strong, bioturbation should be considered, too. The solid phase (OM, iron oxides) in the bioturbation zone is well mixed, which strongly affect OM degradation.

4. I don't understand why the acetoclastic methanogenesis was absent here. Generally acetoclastic methanogenesis dominates in lake sediment and hydrogenotrophic methanogenesis in sea sediment. The two pathways generate different d13C-CH4 and d13-DIC pattern. Diffusion and birrigation will also change this pattern. The authors should prove it.

5. The chemical composition of individual molecules in OM pools can be detected from various state-of-the-art instrumentation including GC-MS, LC-MS/MS, HPLC-MS, NMR, Orbitrap MS, and Fourier transform ion cyclotron resonance (FTICR-MS). By

combining a suite of previously developed thermodynamic theories ( Kleerebezem and Van Loosdrecht, 2010; LaRowe and Van Cappellen, 2011), one can calculate COS. If the results are consistent, the paper method is more convincing.

6. d13C-CH4 in Lake Tantaré Basin A (Fig.3) is very negative ($-107.0$). Is there some explanation?

ref:

Kleerebezem, R., and Van Loosdrecht, M.C.M. (2010). A Generalized Method for Thermodynamic State Analysis of Environmental Systems. Critical Reviews in Environmental Science and Technology 40, 1-54.

Larowe, D.E., and Van Cappellen, P. (2011). Degradation of natural organic matter: A thermodynamic analysis. Geochimica Et Cosmochimica Acta 75, 2030-2042.

---

## Author Comment (AC1) · 13 May 2020

**Reviewer 1:**

**Anonymous Referee #1

**The manuscript is a modification of previous work published by Clayer et al 2018 in Geochimica et Cosmochimica Acta. It builds on the hypothesis that the degradation of organic material under anaerobic conditions has two ultimate sinks – CO2 or CH4, the most reduced and oxidized states of carbon. The relative abundance of these two should therefore provide information on the average oxidation state of the degraded organic material accounting for transport and other sources of CO2. This principal approach has been presented in Clayer et al (2018) GCA. The current paper is very similar to this published work and contains many data that are shared. It was not obvious to me where this work is a significant novel contribution beyond what has also ready been published. The authors use a simple steady state reaction transport model for diffusive and advective transport to determine microbial process rates based on sediment porewater concentration gradients. Furthermore, the isotope composition of DIC is used to improve the mass balance calculations.**

**Below I question the validity of this approach to obtain a meaningful mass balance using the Berg et al model at steady state.**

Thank you for your rigorous and comprehensive comments, they have been very useful to improve the quality of the manuscript. We appreciate the time invested by the reviewer.

**#1 - This manuscript is difficult to understand manuscript and very technical in its description. What makes this manuscript so hard to read and understand is the multitude of R subscripts that are used in the text and the extensive treatment of the methodology in the appendix. The fractional equations are nowhere introduced. Deserves an explanation. In practice, one has to have a table on the side to look up what reaction a particular R subscript refers to and, in addition, know all the notations from Clayer et al (2018) to follow this work.**

We agree that the manuscript is very technical, which makes it harder to follow. In order to rigorously distinguish between net reaction rates, provided by PROFILE, and effective (or gross) reaction rates, we need these numerous R notations. We have now better introduced the effective reaction rates $R_i$. They were only introduced in Table 1 captions, tin the original manuscript. We also better describe the term $R_{net}^{Ox}$. Note also that we have clarified the description of reaction r1 (see response to comment #9)

Note that all notations are described in the manuscript, there is no need to have notations from Clayer et al., 2018.

L. 138 now reads

"The main reactions retained in this study to describe carbon cycling in the sediments of the two lake basins are shown in Table 1. $R_i$ and $\alpha_i$ denote, respectively, the effective (or gross) reaction rate and the carbon isotopic fractionation factor associated with each reaction ri (Table 1)."

Note also that each net reaction rates are explicitly defined.

L.121 "$R_{net}^{solute}$ (in mol cm$^{-3}$ of wet sediment s$^{-1}$) is the solute net production rate (or consumption rate if $R_{net}^{solute}$ is negative)"

L. "the net rate of CH$_4$ production, $R_{net}^{CH_4}$"

L. "The net rate of DIC production, $R_{net}^{DIC}$,"

L. 153 "$R_{net}^{Ox}$ is the net reaction rate of all relevant oxidants consumption, i.e., O$_2$, Fe(III) and SO$_4^{2-}$ only because NO$_3^-$ and Mn(IV) are negligible (see above)." And there after (see also response to comment #15 below).

**#2 - Still, to follow the conclusions becomes increasingly confusing as one reads along, until one is either lost or exhausted. In the current form, the manuscript cannot be digested. I recommend that the authors outline the hypothesis, mathematically, in the materials and methods section, of how their methodology allows them to get at the oxidation state of oxidized organic matter. In the current form, the reader has to wade through too much text to get to this most interesting point of the manuscript. This paper requires a much better didactic approach to get methods and goals across and the authors get sidetracked in many details that make it hard to follow their ultimate goal. It is, in its current form, not streamlined enough and requires very significant rewriting and restructuring to make the approach more understandable and possible to evaluate critically. At present, I cannot evaluate the quality of the manuscript, but am left in doubt about its novelty given the similarity to the 2018 GCA paper. While the fundamental goal, to arrive at the oxidation state of metabolizable organic material, is of some significance, the presentation of the approach is not well developed and can be improved considerably. Data and basic approach (although I did get lost in the complicated d13C treatment) are, in principle feasible, but overall I am concerned that the instrumental and modelling analytical uncertainty is too great to pin down the COS sufficiently (although an error is given).**

We have reorganized the method and discussion sections to better describe the approach and outline the hypothesis mathematically (see our response to comment #3). We have also modified the abstract, introduction and conclusions for consistency and to highlight to novelty compared to the 2018 GCA paper.

L. 13-24 now read:

[revised manuscript text omitted]

**#3 -The authors provide statistical data to support their assertion, but it was not possible for me, based on the complicated description, to relate the outcome of these tests to the goal of the manuscript, i.e., the original oxidation of the degrading organic material. The authors must make sure, in a succinct and understandable and not too wordy fashion, how their methodology allows them to pin this value down sufficiently. Remove as much as possible reiterations of what has already been said and discussed in detail in Clayer et al 2018 GCA and restrict this paper to the novel information.**

Thank you for a constructive comment. In consequence, we have (i) clarified the novel aspects (see response to comment above), and (ii) better described the modelling and COS estimation approaches, (iii) edited the conclusions (see also our response to comment #4) and (iv) simplified the discussion (See also our response to comment #4 and #5).

As a consequence of including Eq. 9 below, Eq. 11 and 12 were removed which simplified section 4.2. COS values displayed in Table 4 are directly calculated with Eq. 9.

The description of the approach now reads:

[revised manuscript text omitted]

**#4 - A lot of the discussion about the CH4 isotopes are not really part of the goal of this paper. This should be separated.**

Agreed, we simplified and moved the text L. 305-317 to section 3.4 to streamline the discussion. Note however that the discussion related to the importance of hydrogenotrophy in section 4.1 is kept and its implication is now emphasized in the Conclusions (see below).

L. 305-317 now, in section 3.4, read:

"The sharp upward depletion in $^{13}$C-CH$_4$ leading to a minimum $\delta^{13}$C-CH$_4$ value at 2.5 cm depth in Lake Tantaré Basin A sediments (Fig. 3a) was unanticipated since it occurs in the methanotrophic zone, i.e., where the remaining CH$_4$ is expected to be $^{13}$C-enriched as a result of CH$_4$ oxidation. Marked $^{13}$C-CH$_4$ depletions at the base of the sulfate-methane transition zone, where CH$_4$ is consumed via SO$_4^{2-}$ reduction, have often been observed in marine sediments (Burdige et al., 2017 and references therein). Such features are generally attributed to the production of CH$_4$ by hydrogenotrophy from the $^{13}$C-depleted DIC resulting from the anaerobic CH$_4$ oxidation, a process referred to as intertwined methanotrophy and hydrogenotrophy (e.g., Borowski et al., 1997; Burdige et al., 2017; Pohlman et al., 2008). Here the modelled $\delta^{13}$C-CH$_4$ profile captured the minimum in $\delta^{13}$C-CH$_4$ in the Z$_1$ by simply assuming concomitant hydrogenotrophy and methanotrophy in this zone and an upward-increasing $\alpha_4$ value from 1.085 in the Z$_3$ to 1.094 in the Z$_1$ (section S2.2.1 of the SI). A small variation with sediment depth in the fractionation factor $\alpha_4$ is arguably possible since its value depends on the types of microorganisms producing CH$_4$ (Conrad, 2005)."

L. 444–447 now read:

"Our results show that fermentation and methanogenesis represent about 50% and 100% of OM mineralization in the top 25 cm of the sediments at the hypolimnetic sites in Lake Tantaré Basin A and Bédard, respectively, that methane is produced only by hydrogenotrophy and fermentation substrates have a negative COS at these two sites. The association of hydrogenotrophy with the fermentation of reduced OM (COS < -0.9; implying that labile compounds are depleted) in the studied

lake sediments is consistent with the fact that hydrogenotrophy becomes increasingly important when labile OM is depleted (Chasar et al., 2000; Hornibrook et al., 2000; Whiticar et al., 1986)."

**#5 - There are a few assumptions whose impact I don't understand or that are difficult to assess, e.g., that there are no anaerobic reoxidation reactions for sulfide; elemental sulfur with FeOOH. The paper does not lost O2 uptake rates for the oxic part of the year, which is an important constraint on the 'background CO2 levels in the buried porewaters. The paper does not constrain oxygen penetration depths or the importance of bioturbation processes for DIC levels, and does not show O2 microelectrode profiles, which would be necessary to constrain the inorganic oxidative processes. Therefore the constraints for the diagenetic system, e.g., by having total O2 uptake rates are far and few. In principle, non-steady state reaction transport modelling with a much more advanced model are necessary to tackle this question, if it is possible at all.**

$O_2$ microprofiles were not measured for this study, but Couture et al. (2016) reported $O_2$ micro-profile measured previously in the sediment of Lake Tantaré Basin A. These microprofiles were the basis of our estimation of $R_{net}^{O_2}$, see L. 129-132. For the majority of the other study sites, the bottom waters were anoxic ($O_2$ <0.1 mg L-1), thus $O_2$ uptake was negligible. Furthermore, we acknowledge in section 4.3 (L. 406-442), the fact that bioirrigation, $O_2$ uptake and misattribution of DIC production can involve uncertainty in the COS estimation for Lake Tantaré Basin A. However, these factors are much less prominent, if not absent, for the other seasonally anoxic lake basins. We believe that the strength of our demonstration resides is the consistency among the COS estimations reported for the seasonally anoxic basins.

We also agree that the concentration profiles presented here were collected in October and are thus the result of "**background CO2 levels in the buried porewaters**", and of changing conditions at the SWI. Although our approach does not enable to resolve all aspect of the complex OM degradation cycling, e.g., explaining the magnitude of all the fluxes involved at a given depth, it allows us to estimate process rates in a given sediment zone, independently of the background concentrations.

Note that the term $R_{net}^{Ox}$ takes into account anaerobic reoxidation reactions for sulfide; elemental sulfur with FeOOH. Admittedly, this point was not stated clearly enough. See our response to comment #6 below.

We also clarify the importance of bioturbation. L. 122-123 now reads:

"considering steady state and negligible solute transport by bioturbation and advection. The validity of these assumptions has been previously demonstrated for the study sites (Couture et al., 2008; Couture et al., 2010; Clayer et al., 2016)."

Regarding non-steady state reaction transport modelling, see also our response to comment #17.

**#6 - Another curious observation is the omission of NO3 dynamics as part of O2 consumption by nitrification; also here there are no constraints on the system concerning NH4+ to accompany C mineralization dynamics.**

**The model also ignores O2 consumption due to Fe oxidation, but curiously the authors choose to include instead sulfide oxidation with O2 and FeOOH.**

Our approach, admittedly not stated clearly enough in the original manuscript, has considered all oxidants. Some of them (NO3 and Mn(IV)) have been shown previously to be negligible because of their low content in the sediment and porewaters (Clayer et al., 2016). Secondary redox reaction as O2 consumption due to Fe(II) oxidation are also taken into account.

To better appreciate these points, we modified section 2.3 (L. 153) as follows:

"$R_{net}^{Ox}$ is the net reaction rate of all relevant oxidants consumption, i.e., $O_2$, Fe(III) and $SO_4^{2-}$ only because $NO_3^-$ and Mn(IV) are negligible (see above). For simplicity, $R_{net}^{Ox}$ is expressed in equivalent moles of $O_2$ consumption rate, taking into account that $SO_4^{2-}$ and Fe(III) have twice and one quarter the oxidizing capacity of $O_2$, respectively. In practice, the value of $R_{net}^{Ox}$ was calculated by adding those of $R_{net}^{O_2}$, $\frac{1}{4}R_{net}^{Fe(III)}$ and $2R_{net}^{SO_4^{2-}}$ where $R_{net}^{O_2}$, $R_{net}^{Fe(III)}$ and $R_{net}^{SO_4^{2-}}$ were estimated with PROFILE. In this calculation, we assumed that all dissolved Fe is in the form of Fe(II), and that the rate of Fe(II) consumption through reactions r7 is negligible compared to those associated with reactions r5 and r6. Under these conditions, $R_{net}^{Fe(III)} = -R_{net}^{Fe}$. It should be noted that using $R_{net}^{O_2}$, $-R_{net}^{Fe}$ and $R_{net}^{SO_4^{2-}}$ to calculate $R_{net}^{Ox}$, we indirectly take into account the re-oxidation of reduced S and Fe(II), respectively, to $SO_4^{2-}$ and Fe(III) by $O_2$. Indeed, with this procedure, we underestimate the terms $\frac{1}{4}R_{net}^{Fe(III)}$ and $2R_{net}^{SO_4^{2-}}$ because re-oxidation reactions are ignored, but we overestimate by the same amount the term $R_{net}^{O_2}$. In other words, omission of these re-oxidation reactions affects only the relative consumption rates of individual oxidants and not the value of $R_{net}^{Ox}$, which is of interest here."

**#7 - It seems very hard to see how the boundary conditions can be reasonably constrained to continue with the approach used by the authors.**

As boundary conditions, we use for each solute their measured concentrations at top and bottom of the concentration profiles (as stated L. 128-129). The model assumes steady-state, and we now mention that this assumption is valid (see below). Under steady-state conditions, the concentrations of solutes at top and bottom of their profiles should not vary with time.

L. 122-123 "considering steady state and negligible solute transport by bioturbation and advection. The validity of these assumptions has been previously demonstrated for the study sites (Couture et al., 2008; Couture et al., 2010; Clayer et al., 2016)."

**#8 - Line comments: Line 139: There are more products than acetate CO2 and CH4 and H2: Formate, propionate, isopropionate, lactate, butyrate, isobutyrate, pyruvate, succinate, etc.; The sum of the latter can be as high as 30% of the total VFA. Ok, acetate is low, but why should it not, if it is consumed by terminal oxidizers? Line 191: No good explanation for the low acetate concentrations?**

We agree, OM degradation occurs and produces other VFA. But eventually, to produce CH4, it will be degraded to acetate and/or CO2 and/or H2 (see Conrad 1999). Reaction r1 is a representation of the fermentation reaction, considering all VFA is out of the scope of this study.

Note that when measuring ions with ion chromatography (L. 109-111), we also looked for VFA, but those were under detection limit (not mentioned in the manuscript).

Besides, we provide evidence that acetoclasty is negligible. Acetate is just not an important degradation product in these sediments. As stated L. 284-287:

"hydrogenotrophy becomes an increasingly important $CH_4$ production pathway: i) when labile OM is depleted (Chasar et al., 2000; Hornibrook et al., 2000; Whiticar et al., 1986), ii) with increasing sediment/soil depth (Conrad et al., 2009; Hornibrook et al., 1997), or iii) with decreasing rates of primary production in aquatic environments (Galand et al., 2010; Wand et al., 2006)"

We also replaced l. 269-279:

"Modeled $\delta^{13}C$ profiles were considered acceptable only when they fell within one standard deviation of the measured $\delta^{13}C$ profiles (grey area fills in Fig. 4). Acceptable modeled $\delta^{13}C$ profiles were obtained only when methanogenesis was 100% hydrogenotrophic, i.e., when $R_3=0$ (see section S2.2.2.1)."

**L.155 Profile underestimates the oxidation rate because it is a fit of a net rate, not a gross rate, e.g., cryptic cycling leads to CO2 production by sulfate reduction in the absence of a curvature in the gradient.**

Agreed, PROFILE provides net reaction rates. This is now clarified as:

"$R_{net}^{Ox}$ is the net reaction rate of all relevant oxidants consumption, i.e., $O_2$, Fe(III) and $SO_4^{2-}$ only because $NO_3^-$ and Mn(IV) are negligible (see above). For simplicity, $R_{net}^{Ox}$ is expressed in equivalent moles of $O_2$ consumption rate, taking into account that $SO_4^{2-}$ and Fe(III) have twice and one quarter the oxidizing capacity of $O_2$, respectively. In practice, the value of $R_{net}^{Ox}$ was calculated by adding those of $R_{net}^{O_2}, \frac{1}{4}R_{net}^{Fe(III)}$ and $2R_{net}^{SO_4^{2-}}$ where $R_{net}^{O_2}$, $R_{net}^{Fe(III)}$ and $R_{net}^{SO_4^{2-}}$ were estimated with PROFILE. In this calculation, we assumed that all dissolved Fe is in the form of Fe(II), and that the rate of Fe(II) consumption through reactions r7 is negligible compared to those associated with reactions r5 and r6. Under these conditions, $R_{net}^{Fe(III)} = -R_{net}^{Fe}$. It should be noted that using $R_{net}^{O_2}$, $-R_{net}^{Fe}$ and $R_{net}^{SO_4^{2-}}$ to calculate $R_{net}^{Ox}$, we indirectly take into account the re-oxidation of reduced S and Fe(II), respectively, to $SO_4^{2-}$ and Fe(III) by $O_2$. Indeed, with this procedure, we underestimate the terms $\frac{1}{4}R_{net}^{Fe(III)}$ and $2R_{net}^{SO_4^{2-}}$ because re-oxidation reactions are ignored, but we overestimate by the same amount the term $R_{net}^{O_2}$. In other words, omission of these re-oxidation reactions affects only the relative consumption rates of individual oxidants and not the value of $R_{net}^{Ox}$, which is of interest here."

**#9 - Line 281 Equation 9 has not been introduced previously. I don't get those fractions.**

As stated in the text, Equation 9 is a simplification of Reaction r1, where x = $\nu_1$. Reaction r1 is displayed in Table 1 and explained in section 2.3. For further clarification, the description of r1 now reads (L. 138):

"Once oxidants are depleted, fermentation of metabolizable OM of general formula $C_xH_yO_z$ can yield acetate, $CO_2$ and $H_2$ (r1). The coefficient $\nu_1$ in r1 constrains the relative contribution of acetoclasty and hydrogenotrophy."

**#10 - Line 322: This sentence is confusing, why should hydrogenotrophy produce DIC coupled to fermentation? Only fermentation r1 may produce CO2.**

Agreed, DIC can only be produced by fermentation and not by hydrogenotrophy. We now clarify as follows:

"This high ratio indicates that DIC was not produced by fermentation (r1) alone in the $Z_2$ of this lake. Indeed, methanogenesis through the coupling of r1 and r4 yields a $R_1/R_4$ ratio of 2 if the fermenting substrate is carbohydrates (COS of 0) and lower than 2 if the fermenting substrate has a negative COS value."

**#11 - L.330: To avoid confusion, a carbon mineralization process leads to the formation of a mineral acid, e.g., carbonic acid. Neither fermentation nor methanogenesis can therefore be called mineralization processes. They are carbon degradation/decomposition processes.**

We were not aware of this definition for the term "mineralization" since this term is widely used to refer to organic matter degradation/decomposition processes (i.e., transformation of organic matter to mineral molecules as DIC, Phosphate, Ammonium and CH4) including oxidation and fermentation reactions, e.g., Burdige 1991; Larowe and Van Cappellen 2011; Arndt et al., 2013.

To avoid any confusion with other uses of the term "mineralization", we precise in the introduction as follows (L. 50):

"Nonetheless, the performance of these models depends on the correct formulation of the complete OM mineralization reactions, e.g., OM decomposition to DIC, phosphate, ammonium and $CH_4$ through oxidation and fermentation reactions (Burdige 1991), particularly in terms of the metabolizable organic compounds involved."

**#12 - Line 332: Please change your terminology Methanogenesis by hydrogenotrophy cannot lead to CO2 formation**

Agreed, we clarify as follows:

"Indeed, the sum of the rates of $CH_4$ production ($\Sigma R_4$), DIC production due to fermentation associated with $CH_4$ formation ($\Sigma R_1 - \Sigma R_4$) and HMW OM partial fermentation ($\Sigma R_2$)"

**#13 - Line 337-340 This conclusion cannot truly be validated with the approach used here.**

We agree that the text was lacking some clarity related to this conclusion. We now believe that our conclusion here is robust given our clarifications related to comment #XX regarding the use of the d13C data to constrain reaction rates. We also precise our point as follows:

"The inclusion of $\delta^{13}C$ data in the present modeling study thus allowed to better constrain the effective rates of $CH_4$ production ($R_4$). Indeed, a value of $R_4 = 119$ fmol $cm^{-3}s^{-1}$ was required in Eq. (7) to produce an acceptable $\delta^{13}C$-$CH_4$ profile (Table 3 and Fig. S3)."

**#14 - Line 353-355 Again, the authors make the mistake of modelling net concentration profiles to extract information on gross rates. The H2 production and CO2 production rates by cryptic cycling are not reflected in curvatures of concentration gradients, these only represented the net effect.**

We now clarify the text as follows:

"The progressive downward increases in dissolved Fe and $SO_4^{2-}$ (Fig. 2e, f, m and n) below ~5 cm depth and decrease in $\Sigma S(-II)$ (Fig. 2n) observed in the porewaters suggest a net production of $H_2$ from r8 in both lakes. However, in the $Z_1$ and $Z_2$ of Lake Tantaré Basin A, the rate of solid Fe(III) reduction ($<3$ fmol cm$^{-3}$ s$^{-1}$; calculated from Liu et al. 2015) is much lower than that required from r8 (i.e., 1 to 2 times the additional $H_2$ production of $4R_4 - 2R_1$; 70–424 fmol cm$^{-3}$ s$^{-1}$) to produce sufficient amounts of $H_2$ to sustain the additional hydrogenotrophy. The net production rates of dissolved Fe ($<10$ fmol cm$^{-3}$ s$^{-1}$) and $SO_4^{2-}$ ($<1$ fmol cm$^{-3}$ s$^{-1}$) and the net consumption rate of $\Sigma S(-II)$ ($<1$ fmol cm$^{-3}$ s$^{-1}$) are also consistent with this assertion (Fig. 2)."

**#15 - A cryptic sulfur cycle is only used to argue for H2 production. Why not CO2 production by sulfate reduction?**

DIC production by sulfate reduction is considered where the value of Rox_net is positive. Admittedly our phrasing was confused. We now clarify as follows:

"$R_{net}^{Ox}$ is the net reaction rate of all relevant oxidants consumption, i.e., $O_2$, Fe(III) and $SO_4^{2-}$ only because $NO_3^-$ and Mn(IV) are negligible (see above). For simplicity, $R_{net}^{Ox}$ is expressed in equivalent moles of $O_2$ consumption rate, taking into account that $SO_4^{2-}$ and Fe(III) have twice and one quarter the oxidizing capacity of $O_2$, respectively. In practice, the value of $R_{net}^{Ox}$ was calculated by adding those of $R_{net}^{O_2}, \frac{1}{4}R_{net}^{Fe(III)}$ and $2R_{net}^{SO_4^{2-}}$ where $R_{net}^{O_2}, R_{net}^{Fe(III)}$ and $R_{net}^{SO_4^{2-}}$ were estimated with PROFILE. In this calculation, we assumed that all dissolved Fe is in the form of Fe(II), and that the rate of Fe(II) consumption through reactions r7 is negligible compared to those associated with reactions r5 and r6. Under these conditions, $R_{net}^{Fe(III)} = -R_{net}^{Fe}$. It should be noted that using $R_{net}^{O_2}, -R_{net}^{Fe}$ and $R_{net}^{SO_4^{2-}}$ to calculate $R_{net}^{Ox}$, we indirectly take into account the re-oxidation of reduced S and Fe(II), respectively, to $SO_4^{2-}$ and Fe(III) by $O_2$. Indeed, with this procedure, we underestimate the terms $\frac{1}{4}R_{net}^{Fe(III)}$ and $2R_{net}^{SO_4^{2-}}$ because re-oxidation reactions are ignored, but we overestimate by the same amount the term $R_{net}^{O_2}$. In other words, omission of these re-oxidation reactions affects only the relative consumption rates of individual oxidants and not the value of $R_{net}^{Ox}$, which is of interest here."

**#16 - The overall problem with the approach is that a balance based on CO2 and CH4 and the OM oxidation state is too poorly constrained. In reality, in addition to a mass balance an independent charge balance should be achieved to constrain the original oxidation state of the OM. The current approach balances the electrons between the mass of methane and total CO2 accounting for diffusive transport. This could likewise be achieved by adjusting the alkalinity.**

Agreed, an independent charge balance would have been very useful, but our current dataset does not enable to perform it. Note, however, that our mass balance is corroborated with the isotopic mass balance which add some robustness to our approach. In addition, the number of sites where our approach yields consistent results (i.e., a COS value < -1.0), especially for the seasonally anoxic sites where oxidation reactions, DIC production through partial fermentation (r2) and bioirrigation

do not prevent the accurate estimation of fermentation reaction rates (see also l. 423 – 442), provide a strong support for the robustness of our approach.

**#17 - I think that the authors use the model the wrong way. It is perfectly fine for comparing rates in the different zones, but it is not possible to balance the inventories in the respective zones with this model. A more sophisticated reaction transport model that accounts for the cumulative amount of DIC formed during burial needs to be used to explain the amount and isotope composition of DIC. The model Profile only captures a snapshot of a concentration distribution, i.e., the steady state, and it does not allow for calculating cumulative effects during burial, which is important for a diagenetic model and for this case to account for the buried amount of DIC from oxic respiration. In addition, the steady state assumption is invalid for most natural cases except for very small distance, e.g., at the micrometer scale where diffusion is extremely fast. A time-dependent model that includes mass accumulation rates must be used here.**

We agree that our dataset only provide a snapshot of the complex OM degradation cycling. The concentration profiles presented here are the result of "**the cumulative amount of DIC formed during burial**", and of changing conditions at the SWI. However, the inverse modelling tools used here with the assumption of steady state, whose validity has been discussed previously several times for the study sites (e.g., Clayer et al., 2016; Couture et al., 2008; Feyte et al., 2012), enables us to obtain the net reaction rates for this snapshot for CH4, DIC and Oxidants (relevant here are O2, Fe(III), and SO4) independently of the background concentrations.

We are confident that the depth distributions of the net reaction rates that we present in Fig. 3g, h, o and p for [CH4] and DIC are robust. Indeed, the statistical F-testing implemented in PROFILE allows to objectively select, among all the possible solutions, the one that gives the simplest rate profile while providing a satisfying explanation of the measured solute concentration profile. Also, as can be seen in Fig. S1, using another inverse modeling code, i.e., Rate Estimation from Concentrations (REC, Lettmann et al., 2012), produces consistent results with those obtained with the code PROFILE.

After having seriously considered the suggestion of the reviewer to use a non-steady state model, we decided to keep our inverse modeling approach since we believe it is reliable as described above. To our knowledge, a non-steady state model is not necessarily better suited to interpret the concentration profiles because it requires a high number of adjustable parameters (e.g., the flux of labile organic carbon, of dissolved oxygen and other oxidants, the rate constants for each reaction of OM degradation) which is not the case for the inverse model.

We do not have the pretention to resolve all aspects of the complex OM degradation cycling, e.g., explaining the magnitude of all the fluxes involved at a given depth. We use the net reaction rates obtained by PROFILE in our isotopic model to constrain the gross rates and estimate process rates in a given sediment zone.

*REFs:*

Clayer, F., Gobeil, C. and Tessier, A.: Rates and pathways of sedimentary organic matter mineralization in two basins of a boreal lake: Emphasis on methanogenesis and methanotrophy: Methane cycling in boreal lake sediments, Limnology and Oceanography, 61(S1), S131–S149, doi:10.1002/lno.10323, 2016.

Conrad, R. (1999). Contribution of hydrogen to methane production and control of hydrogen concentrations in methanogenic soils and sediments. FEMS Microbiology Ecology, 28(3), 193–202. https://doi.org/10.1111/j.1574-6941.1999.tb00575.x

Couture, R.-M., Gobeil, C. and Tessier, A.: Chronology of Atmospheric Deposition of Arsenic Inferred from Reconstructed Sedimentary Records, Environ. Sci. Technol., 42(17), 6508–6513, doi:10.1021/es800818j, 2008.

---

## Author Comment (AC2) · 13 May 2020

Reviewer 2:

Anonymous Referee #2

**The paper addresses an interesting and fundamentally important question: which fraction of sedimentary organic matter is mineralized through methanogenesis. Based on modeling and analyses of data from two lakes, it argues that organic carbon in negative oxidation states is used preferentially and the hydrogenotrophic pathway of methanogenesis dominates. If true, this may have profound implications for modeling the carbon cycle and interpretations of sedimentary signatures of carbon isotopes. Both the dataset and the model go well beyond the level of detail of typical diagenetic studies, which is indeed a requirement for figuring out the important fine details of organic matter mineralization.**

**This important work, however, could be improved in several key areas.**

We are thankful to the reviewer for constructive and rigorous comments. We believe that it helped improve the manuscript.

*Style and clarity:* **The clarity of the narrative deteriorates towards the end of the manuscript. In particular, stating clearly and emphasizing throughout the text the main finding of the work would greatly improve readability. Inferences from modeling of the isotopic profiles could also benefit from a clearer presentation. Key statement such as (Line 265) "practically all CH4 is produced through hydrogenotrophy" are inferred from modeling d13C profiles, but I admit I was rather lost following the description, particularly trying to separate the relative contributions of hydrogenotrophic vs acetoclastic methanogenesis.**

Thank you for a constructive comment. In consequence, we have (i) clarified the novel aspects (see response to next comment), (ii) better described the modelling and COS estimation approaches, (iii) moved and focused L. 305-317 to section 3.4, (iv) streamlined section 4.3 (L. 406-442) and (v) edited the conclusions (see comments below).

The description of the approach now reads:

[revised manuscript text omitted]

***Originality:*** **Much of the work is an update on the results of Clayer et al. 2018. The text should clearly distinguish the novel aspects, especially how (or if) the difference in conclusions is more than just refinement of the numbers from that previous work. For example, a statement on lines 58-60 reads: "Based on the observation that methanogenesis produced CH4 three times faster than CO2 . . .. Clayer et al. (2018) concluded that the fermenting OM had a markedly negative COS value of -1.9". This parallels the statement in the Abstract, which presumably should highlight the results from this work: "we calculate, from CH4 and DIC production rates. . .COS below -0.9". This seems to convey the same information.**

We agree. There is some overlap with the results of Clayer et al., 2018, although new datasets are presented and additional data from published work is re interpreted.

We have modified the abstract, introduction and conclusions (see our response to comments "Conclusions" and "Line 454" for changes in the conclusion) to better highlight the novel aspect of the present study.

L. 13-24 now read:

"To test the validity of this assumption, we modeled using reaction-transport equations vertical profiles of the concentration and isotopic composition ($\delta^{13}C$) of $CH_4$ and DIC in the top 25 cm of the sediment column from two lake basins, one whose hypolimnion is perennially oxygenated and one with seasonal anoxia. Furthermore, we modeled solute porewater profiles reported in the literature for four other seasonally anoxic lake basins. A total of seventeen independent porewater datasets are analysed. $CH_4$ and DIC production rates associated with methanogenesis at the five seasonally anoxic sites collectively show that the fermenting OM has a mean ($\pm$SD) carbon oxidation state (COS) value of $-1.4 \pm 0.3$. This value is much lower than the value of zero expected from carbohydrates fermentation. We conclude that carbohydrates do not adequately represent the fermenting OM in hypolimnetic sediments and propose to include the COS in the formulation of OM fermentation in models applied to lake sediments to better quantify sediment $CH_4$ outflux. This study highlights the potential of mass balancing the products of OM mineralization to characterize labile substrates undergoing fermentation in sediments."

And L. 68-74:

"In this study, the approach described in Clayer et al. (2018), combining concentration and $\delta^{13}C$ inverse modeling, is applied to the two newly acquired datasets. These datasets include centimeter-scale vertical porewater profiles of the concentrations and of the stable carbon isotope ratios ($\delta^{13}C$) of $CH_4$ and dissolved inorganic carbon (DIC), as well as those of the concentrations of EAs from hypolimnetic sediments of two boreal lake basins showing contrasted $O_2$ dynamics: one whose hypolimnion remains perennially oxygenated and the other whose hypolimnion becomes anoxic for several months annually. This procedure enables us to constrain the effective rates of OM mineralization reactions and calculate, using a mass balance equation, the COS of the substrates fermenting in the sediments in these two lake basins. In addition, we modelled solute porewater profiles gathered from the scientific literature or from our data repository for four other seasonally anoxic lake basins to estimate, using the mass balance equation, the COS of the substrates fermenting in these sediments. A total of seventeen independent datasets are analysed to provide additional insight into the COS of the fermenting OM in boreal lakes and the associated mineralization pathways."

***Justifying the inclusion or omission of processes:*** **The coupling with the sulfur cycle seems particularly suspect. The cryptic oxidation of sulfide coupled to iron oxides is used as an important pathway for H2 production. While this reaction is commonly considered (but can be written in various stoichiometries), it is rarely the only reaction that is considered from the complicated network of reactions that comprise the sedimentary Fe and S cycling. Puzzlingly, the modeled SO4 and Fe profiles are not shown (line235). These absolutely need to be shown. The sulfur cycle in this system seems highly unusual. For example (Line 201 and Fig. 2), "SO42- concentrations reach a minimum between SWI and 5 cm depth, and increase below". These highly unusual features need to be discussed. How can SO4 be produced in anoxic sediment? Does oxidation of H2S by Fe(III) somehow proceed faster than sulfate reduction? What about precipitation of iron sulfides? Similarly, precipitation of CaCO3 does not seem to be considered as a CO2 sink, while Line 380 mentions that it had to be considered by the used datasets. Were the saturation indexes negative for the study sites?**

We agree that the description of the reactions was lacking some rigor in section 2.3.

Unraveling the complex Fe and S cycling is, however, out of the scope of this study, we now refer to Couture et al. (2016). Nonetheless, note that these features described at L. 201 are discussed at L. 353-360 which have been modified following the comment of another reviewer as described below. Regarding the precipitation of iron sulfides we now clarify that iron sulfide are currently experiencing dissolution, hence precipitation could be neglected (see below).

As stated L. 144, "the precipitation of carbonates (*can be neglected*) whose saturation index values are negative (SI ≤ −1.5) except for siderite (r7) in Lake Bédard (SI = 0.0 to 0.7 below 10 cm depth)"

Finally, we added the modelled Fe and SO4 profiles into an additional figure in the supplementary information Fig. S3. and refer to it in the text:

[Figure]

"

**Figure S3: Comparison of the modeled (blue lines) and average (n = 3) measured (symbols) concentration profiles of SO₄ (a and c) and Fe (b and d) in Lakes Tantaré Basin A (a–b) and Bédard (c–d). The horizontal dotted line indicates the sediment-water interface. The thick red lines represent the net solute reaction rate ($R_{net}^{solute}$).**"

Regarding reducing Fe and S cycling, it now reads:

"Lastly, sulfide oxidation by iron oxides (r8), which can be a source of $SO_4^{2-}$ and $H_2$ (Clayer et al., 2018; Holmkvist et al., 2011), is also considered. Note that iron sulfide enrichments formed during past decades of elevated atmospheric SO₄ deposition are presently dissolving in Lake Tantaré Basin A (Couture et al., 2016). This process also occurs in the seasonally anoxic Basin B of Lake Tantaré (Couture et al., 2016) and is likely to also occur in Lake Bédard. Hence, other reactions involving reduced S and Fe species, such as pyrite precipitation, are believed to be insignificant for C cycling in the present study and are thus ignored."

L. 353-360 now read:

"The progressive downward increases in dissolved Fe and $SO_4^{2-}$ (Fig. 2e, f, m and n) below ~5 cm depth and decrease in $\Sigma S(-II)$ (Fig. 2n) observed in the porewaters suggest a net production of $H_2$ from r8 in both lakes. However, in the $Z_1$ and $Z_2$ of Lake Tantaré Basin A, the rate of solid Fe(III) reduction (<3 fmol cm⁻³ s⁻¹; calculated from Liu et al. 2015) is much lower than that required from r8 (i.e., 1 to 2 times the additional $H_2$ production of $4R_4 - 2R_1$; 70–424 fmol cm⁻³ s⁻¹) to produce sufficient amounts of $H_2$ to sustain the additional hydrogenotrophy. The net production rates of dissolved Fe (<10 fmol cm⁻³ s⁻¹) and $SO_4^{2-}$ (<1 fmol cm⁻³ s⁻¹) and the net consumption rate of $\Sigma S(-II)$ (<1 fmol cm⁻³ s⁻¹) are also consistent with this assertion (Fig. 2)."

*Discussing implications:* **If the organic matter used in methanogenesis had negative COS, what happened to the rest of the C pool? Is oxidized OM not mineralized? Or is it mineralized preferentially earlier, in the water column? What are the implications, e.g. for burial, signature of OM in rock record, etc.? The statement on line 450 seems to address it somewhat, but the statement is not clear.**

The implications of our study are now better described although we believe it does not influence burial or the signature of OM in rock record since only a very small fraction of the C pool is mineralized (see below).

Statement on line 450 has been clarified as follows:

"We propose that the most labile compounds are mineralized during OM downward migration in the water column and in the uppermost sediment layers leaving mainly reduced organic compounds to fuel methanogenesis in these sediments."

L. 334 the following sentences were added regarding the rest of the C pool:

"Considering the sediment accumulation rate and sediment $C_{org}$ content given in section 2.1, we calculate an average accumulation rate of $C_{org}$ of $4.7\times10^{-11}$ to $1.0\times10^{-10}$ and $2.9\times10^{-11}$ to $7.6\times10^{-10}$ mol C cm$^{-2}$ s$^{-1}$ for lakes Tantaré Basin A and Bédard, respectively. Hence, the total sediment OM degradation rate ($\Sigma R_1 + \Sigma R_2 + \Sigma R_6$) of $1.3\times10^{-12}$ and $1.4\times10^{-12}$ reported in this study for lakes Tantaré Basin A and Bédard, respectively, would involve only 1.2−2.8% and 0.2−4.8% of the total $C_{org}$ deposited. Given that the remaining 95.2−99.8% of the deposited $C_{org}$ is preserved in the sediment, it is not surprising that the sediment $C_{org}$ concentration is constant with depth (Fig. 2)."

**It would also help to discuss how special or typical these lakes are, given that the implications seem to include global extrapolations. For example, diagenesis in Lake Tantare (or is it Lake Bedard? – see below) seems to lack contributions from terminal electron acceptors. How different would this be from a "typical" boreal forest lake?**

To be able to better assess how "typical" our case study lakes are, we added some background information on the sediment OM in Section 2.1. In addition, we included a brief discussion to which degree they are representative of boreal forest lakes.

We added a figure and some information on the sediment OM in section 2.1 as follows:

"The sediment accumulation rates are 4.0–7.3 and 2.4–46.8 mg cm$^{-2}$ yr$^{-1}$ at the deepest sites of Lake Tantaré Basin A and Lake Bédard, respectively (Couture et al., 2010). The relatively constant organic C ($C_{org}$) content (20 ± 2%; Fig. 2b), the elevated $\{C_{org}\}$:$\{N\}$ molar ratio (17 ± 2; Fig. 2b), the $\delta^{13}$C (−29‰; Joshani, 2015) and $\delta^{15}$N (+0.5‰ to −2.5‰; Joshani, 2015) values reported for the sediment OM over the top 30 cm in Lake Tantaré Basin A are typical of terrestrial humic substances (Botrel et al., 2014; Francioso et al., 2005). The $C_{org}$ content (21 ± 2.7%; Fig. 2a) and $\{C_{org}\}$:$\{N\}$ molar ratio (14 ± 1.9; Fig. 2a) reported over the top 30 cm of Lake Bédard sediments show slightly more variation with depth, but are also typical of terrestrial OM. In addition, the $\{C_{org}\}$:$\{S\}$ ratios of both lake basin sediments (50–200) are typical of those reported for soil OM (~125; Buffle, 1988).

[Figure]

Figure 2: Depth profiles of the organic C concentrations and of the C : N molar ratio in sediment cores collected at the deepest sites of Lake Bédard (a) and Lake Tantaré Basin A (b)."

L. 450 – 453 now read:

"The OM in the sediment of the three boreal lakes, as well as their $O_2$ seasonal dynamics, are typical of boreal forest lakes. While Lake Bédard experiences prolonged episodes of extended hypolimnetic anoxia, Lake Tantaré Basin B and Jacks Lake show more moderate seasonal anoxia, where some years the hypolimnion of Lake Tantaré Basin B is only hypoxic (Clayer et al., 2016; Carignan et al., 1991). Hence, the selective mineralization of OM described by Clayer et al. (2018), involving that the most labile compounds are mineralized during OM downward migration in the water column and at the sediment surface leaving mainly reduced organic compounds to fuel methanogenesis in the sediments, likely applies to a large portion of boreal lakes."

***Other criticisms and suggestions:***

***Conclusions:*** **"fermentation and methanogenesis represent. . .100% of OM mineralization . . . in Lake Tantare" – Methanogenesis can be fermentation. More importantly, why are there no contributions from terminal electron acceptors? Is it really 100%? Confusingly, Fig. 2 shows that sulfate reduction is clearly active in Lake Tantare, whereas contributions of terminal electron acceptors are likely smaller in Lake Bedard.**

We apologize, this is a mistake, it should the other way around. The correct sentence now reads:

"Our results show that fermentation and methanogenesis represent about 50% and 100% of OM mineralization in the top 25 cm of the sediments at the hypolimnetic sites in Lake Tantaré Basin A and Bédard, respectively"

**One of the main results seems to be expressed by Eq. 15. Given the range of COS values (-1.4+- 0.3), it might be helpful to state the range in the stoichiometric coefficients explicitly.**

Agreed, we have added a sentence L. 460

"Introducing the average COS values reported in this study ($-1.4 \pm 0.3$) into Eq. 15, the coefficients $a$ and $b$ would take values of $2.7\pm0.15$ and $0.65\pm0.125$, respectively, and the $CH_4$ and $CO_2$ stoichiometric coefficients would be $0.68\pm0.04$ and $0.32\pm0.04$, respectively."

**Line 284: "i) when labile OM is depleted, ii) with increasing sediment depth" – aren't these two statements in practice the same?**

The first statement refers to OM depletion across time, while the other is across space.

**Line 454: "misestimating CH4 and CO4 production" – not sure what this means. Underestimating the amounts? But early diagenetic models generally work okay and can reproduce measured profiles. Are the differences small enough that they are within uncertainties?**

Early diagenetic models are rarely validated against both CH4 and DIC profiles at the same time. Below we also better describe how significant our findings could be for CH4 sediment fluxes and oxidant consumption rates.

To better clarify, we modified L. 454 as follows:

"Hence, the current representation of the fermenting OM, i.e., $CH_2O$, in process-based biogeochemical models entails a significant risk of underestimating sedimentary $CH_4$ production and release to the bottom water and, to a certain extent, of its evasion to the atmosphere under transient environmental scenarios."

Added the following text L. 460:

"Introducing the average COS values reported in this study ($-1.4 \pm 0.3$) into Eq. 15, the coefficients $a$ and $b$ would take values of $2.7\pm0.15$ and $0.65\pm0.125$, respectively, and the $CH_4$ and $CO_2$ stoichiometric coefficients would be $0.68\pm0.04$ and $0.32\pm0.04$, respectively. Note that the same stoichiometric formulation would be obtained for acetoclastic methanogenesis. Under these conditions, fermentation (r1) coupled to methanogenesis (r4) yields $2.2\pm0.4$ times more $CH_4$ than DIC for the studied lake sediments. Ignoring the implications of the present study regarding the COS of the fermenting OM could lead to the underestimation of $CH_4$ sediment outflux or of the rate of oxidant consumption required to mitigate this efflux by a factor of up to 2.6."

***REFs:***

Botrel, M., Gregory-Eaves, I. & Maranger, R. Defining drivers of nitrogen stable isotopes (δ15N) of surface sediments in temperate lakes. Journal of Paleolimnology 52, 419-433 (2014).

Clayer, F., Gobeil, C. and Tessier, A.: Rates and pathways of sedimentary organic matter mineralization in two basins of a boreal lake: Emphasis on methanogenesis and methanotrophy: Methane cycling in boreal lake sediments, Limnology and Oceanography, 61(S1), S131–S149, doi:10.1002/lno.10323, 2016.

Conrad, R. (1999). Contribution of hydrogen to methane production and control of hydrogen concentrations in methanogenic soils and sediments. FEMS Microbiology Ecology, 28(3), 193–202. https://doi.org/10.1111/j.1574-6941.1999.tb00575.x

Couture, R.-M., Gobeil, C. and Tessier, A.: Chronology of Atmospheric Deposition of Arsenic Inferred from Reconstructed Sedimentary Records, Environ. Sci. Technol., 42(17), 6508–6513, doi:10.1021/es800818j, 2008.

Francioso, O., Montecchio, D., Gioacchini, P., and Ciavatta, C. Thermal analysis (TG–DTA) and isotopic characterization (13C–15N) of humic acids from different origins (2005) Applied Geochemistry 20(3), 537-544

---

## Author Comment (AC3) · 13 May 2020

Reviewer 3:

**Anonymous Referee #3

**In this paper, Clayer et al. found the average carbon oxidation state (COS) is negative COS values by modelling solute pore-water profiles. They concluded that carbohydrates do not adequately represent the fermenting OM and that the COS should be included in the formulation of OM fermentation in models. It is an interesting work and the results can guide new biogeochemical model for OM degradation. However, the manuscript needs substantial improvement of the presentation before it can be recommended for publication.**

We are thankful to the reviewer for constructive and rigorous comments. We believe that it helped improve the manuscript.

**The main issues is the lack of the OM and mobile labile information. There are no data for the deposition/sedimentation rate of OM, the chemical composition of OM (C,H,O,N,S,P,..), d13C distribution of OM et al. The results of COS from modelling solute pore-water profiles have not been validated. Even in the solute model there are too many fitting parameters and the conclusion is not convincing.**

Agreed, background information on OM was lacking.

As we understand it, the reviewer would also have appreciated to see chemical composition data on single organic compounds that corroborate our COS estimations. However, we do not dispose of such analytical methods nor of any additional samples to perform these analyses. We agree that it could have been an interesting complement. However, we believe that the strength of our demonstration resides is the consistency among the COS estimations reported for the seasonally anoxic basins. See also our response to your comment #5 below.

Regarding the solute model, we now have re-organized the methods description to better describe our approach in a convincing way. While the robustness of the net reaction rates obtained with PROFILE is clearly highlighted L. 226-232. Extracts of the method sections now reads

"Considering the net reaction rates obtained by inverse modelling, a realistic range of values can be given for each of the effective reaction rates $R_i$ in each depth interval, as determined by PROFILE, using the general equations described below (Eqs. 3, 4 and 5). The detailed calculations for each $R_i$ at both study sites are described in section S2.

(…)

Once the range of values have been determined for each of the effective rates $R_i$ (see Table S2), they can be used in another reaction-transport equation to model the $\delta^{13}C$ profiles of $CH_4$ and DIC. Only sets of $R_i$ values that yield acceptable modeled $\delta^{13}C$ profiles, i.e., which fall within one standard deviation of the measured $\delta^{13}C$ profiles (grey area fills in Fig. 4), were kept for COS calculation below (section 2.8). The $\delta^{13}C$ modeling procedure is summarized below and described in detail in Section S.2. This procedure takes into account the effect of diffusion, bioirrigation (in Lake Tantaré Basin A) and the isotopic fractionation effect of each reaction ri.

(…)

**2.8 COS calculation**

Considering the complete fermentation of metabolizable OM of general formula $C_xH_yO_z$, and making two assumptions, described below for clarity, the COS of the fermenting molecule is given by (combining Eq. S8 and S15; see Section S2 for details):

$$COS = -4 \left( \frac{R_{net}^{CH_4} - R_{net}^{DIC} - R_{net}^{Ox} + R_2}{R_{net}^{CH_4} + R_{net}^{DIC} + (1 - \chi_M)R_{net}^{Ox} - R_2} \right) \tag{9}$$

where $\chi_M$ is the fraction of oxidants consumed by methanotrophy. Equation (9) is only valid if i) r1 is the only source of substrates for hydrogenotrophy and acetoclasty (this assumption is discussed in Section 4.2 below); and that ii) siderite precipitation (r7) is negligible (Saturation Index for siderite are negative except below 10 cm depth in the sediment of Lake Bédard, this case is considered in Section S2.1.2.2). With values of $R_{net}^{CH_4}$ and $R_{net}^{Ox}$ obtained from PROFILE (section 2.4), values of $R_1$, $\chi_H$ and $\chi_M$ constrained by $\delta^{13}C$ modeling (section 2.7), Eq. (9) can be used to calculate the COS of the fermenting molecule."

We added a figure and some information on the sediment OM as follows:

"The sediment accumulation rates are 4.0–7.3 and 2.4–46.8 mg cm$^{-2}$ yr$^{-1}$ at the deepest sites of Lake Tantaré Basin A and Lake Bédard, respectively (Couture et al., 2010). The relatively constant organic C ($C_{org}$) content ($20 \pm 2\%$; Fig. 2b), the elevated $\{C_{org}\}:\{N\}$ molar ratio ($17 \pm 2$; Fig. 2b), the $\delta^{13}C$ ($-29$‰; Joshani, 2015) and $\delta^{15}N$ ($+0.5$‰ to $-2.5$‰; Joshani, 2015) values reported for the sediment OM over the top 30 cm in Lake Tantaré Basin A are typical of terrestrial humic substances (Botrel et al., 2014; Francioso et al., 2005). The $C_{org}$ content ($21 \pm 2.7\%$; Fig. 2a) and $\{C_{org}\}:\{N\}$ molar ratio ($14 \pm 1.9$; Fig. 2a) reported over the top 30 cm of Lake Bédard sediments show slightly more variation with depth, but are also typical of terrestrial OM. In addition, the $\{C_{org}\}:\{S\}$ ratios of both lake basin sediments (50–200) are typical of those reported for soil OM ($\sim 125$; Buffle, 1988).

[Figure]

Figure 2: Depth profiles of the organic C concentrations and of the C : N molar ratio in sediment cores collected at the deepest sites of Lake Bédard (a) and Lake Tantaré Basin A (b)."

**Here are some details:**

**1. Reactions: Since the reactions the precipitation of siderite (r7) and sulfide oxidation by iron oxides (r8) were taken into account, the pyrite formation by Fe2+ and H2S should be considered,too. The hydrogen H2 in eq.(r8) is usually consumed easily by sulphate reducer bacteria rather than CO2 reduction. The authors used general oxidant instead of O2,Fe(III) and SO4, which could have different oxidation rates, especially for CH4 oxidation (r5).**

We agree that the description of the reactions was lacking some rigor in section 2.3. Regarding reducing Fe and S cycling, it now reads:

"Lastly, sulfide oxidation by iron oxides (r8), which can be a source of $SO_4^{2-}$ and $H_2$ (Clayer et al., 2018; Holmkvist et al., 2011), is also considered. Note that iron sulfide enrichments formed during past decades of elevated atmospheric $SO_4$ deposition are presently dissolving in Lake Tantaré Basin A (Couture et al., 2016). This process also occurs in the seasonally anoxic Basin B of Lake Tantaré (Couture et al., 2016) and is likely to also occur in Lake Bédard. Hence, other reactions involving reduced S and Fe species, such as pyrite precipitation, are believed to be insignificant for C cycling in the present study and are thus ignored."

Concerning the consumption of H2 by sulphate reducer, it possibly occurs in the top 5 cm in Lake Tantaré Basin A, but considering the low SO4 concentrations, this process is likely negligible. Note that we have added a figure in the supplementary information showing modelled SO4 profiles and reaction rates. This figure shows net SO4 production below 5-7 cm depth at a very low rate.

Note that the various oxidation state of the oxidants are taken into account as now stressed L. 153

"$R_{net}^{Ox}$ is the net reaction rate of all relevant oxidants consumption, i.e., $O_2$, Fe(III) and $SO_4^{2-}$ only because $NO_3^-$ and Mn(IV) are negligible (see above). For simplicity, $R_{net}^{Ox}$ is expressed in equivalent moles of $O_2$ consumption rate, taking into account that $SO_4^{2-}$ and Fe(III) have twice and one quarter the oxidizing capacity of $O_2$, respectively. In practice, the value of $R_{net}^{Ox}$ was calculated by adding those of $R_{net}^{O_2}$, $\frac{1}{4}R_{net}^{Fe(III)}$ and $2R_{net}^{SO_4^{2-}}$ where $R_{net}^{O_2}$, $R_{net}^{Fe(III)}$ and $R_{net}^{SO_4^{2-}}$ were estimated with PROFILE. In this calculation, we assumed that all dissolved Fe is in the form of Fe(II), and that the rate of Fe(II) consumption through reactions r7 is negligible compared to those associated with reactions r5 and r6. Under these conditions, $R_{net}^{Fe(III)} = -R_{net}^{Fe}$. It should be noted that using $R_{net}^{O_2}$, $-R_{net}^{Fe}$ and $R_{net}^{SO_4^{2-}}$ to calculate $R_{net}^{Ox}$, we indirectly take into account the re-oxidation of reduced S and Fe(II), respectively, to $SO_4^{2-}$ and Fe(III) by $O_2$. Indeed, with this procedure, we underestimate the terms $\frac{1}{4}R_{net}^{Fe(III)}$ and $2R_{net}^{SO_4^{2-}}$ because re-oxidation reactions are ignored, but we overestimate by the same amount the term $R_{net}^{O_2}$. In other words, omission of these re-oxidation reactions affects only the relative consumption rates of individual oxidants and not the value of $R_{net}^{Ox}$, which is of interest here."

New figure:

[Figure]

"

**Figure S3: Comparison of the modeled (blue lines) and average (n = 3) measured (symbols) concentration profiles of SO4 (a and c) and Fe (b and d) in Lakes Tantaré Basin A (a–b) and Bédard (c–d). The horizontal dotted line indicates the sediment-water interface. The thick red lines represent the net solute reaction rate ($R_{net}^{solute}$)."**

**2. Rate calculation: The reaction rate were calculated by computer code PROFILE. This rates obtained from PROFILE were very rough. It is better to use reactiontransport model to calculate the rate by considering OM deposition and degradation.**

We agree with the reviewer that a limitation of inverse modeling methods is that the predicted rate profiles may not be unique. However, we are confident that the depth distributions of the net reaction rates that we present in Fig. 3g, h, o and p for [CH4] and DIC are robust. Indeed, as stated L. 226-232 the statistical F-testing implemented in PROFILE allows to objectively select, among all the possible solutions, the one that gives the simplest rate profile while providing a satisfying explanation of the averaged solute concentration profile. Also, as can be seen in Fig. S1, using another inverse modeling code, i.e., Rate Estimation from Concentrations (REC, Lettmann et al., 2012), produces consistent results with those obtained with the code PROFILE. Moreover, the values of the net rates are of similar magnitude. Note that REC uses the Tikhonov regularization technique. This statistical method implies the adjustment of one discrete parameter (i.e., the smoothing parameter $\lambda$) and, in contrast to PROFILE, does not suggest a given number of zones.

After having seriously considered the suggestion of the reviewer, we decided to keep our inverse modeling approach since we believe it is reliable as described above. To our knowledge, a non-steady state model is not necessarily better suited to interpret the concentration profiles because it

requires a high number of adjustable parameters (e.g., the flux of labile organic carbon and of dissolved oxygen and other oxidants, the rate constants for each reaction of OM degradation) which is not the case for the inverse model. In addition, using a forward model would imply making additional subjective choices regarding the rate expressions and boundary conditions, e.g., parametrizing the O2 sediment flux.

**3. The bioirrigation term was shown in the equation (2) but the bioirrigation depth and coefficient were not clear. How does the bioirrigation affect COS estimatation was also not clear. General once bioirrigation is strong, bioturbation should be considered, too. The solid phase (OM, iron oxides) in the bioturbation zone is well mixed, which strongly affect OM degradation.**

Agreed, there was some information lacking regarding biological processes. Bioturbation has been shown previously to be negligible compared to diffusion or biorrigation (e.g, Clayer et al., 2016; Couture et al., 2008) This is now fixed as follows (in section 2.3):

L. 124 "The values of $\alpha_{Irrigation}$ in Lake Tantaré Basin A were calculated as in Clayer et al. (2016), considering that it varies linearly from $\alpha_{0\_Irrigation}$ at the SWI (calculated according to Boudreau 1984 based on an inventory of benthic animals Hare et al., 1994) to 0 at 10 cm depth (the maximum depth at which chironomids are found in lake sediments; Matisoff and Wang 1998), and were assumed to be 0 in Lake Bédard since its bottom water was anoxic (Fig. 1)."

L. 122-123 "considering steady state and negligible solute transport by bioturbation and advection. The validity of these assumptions has been previously demonstrated for the study sites (Couture et al., 2008; Couture et al., 2010; Clayer et al., 2016)."

To describe the sensitivity of COS values to the bioirrigation term we added the following sentence L. 422:

"Even if reaction rates are sensitive the value of the bioirrigation coefficient (Clayer et al., 2016), additional simulations show that changing the bioirrigation coefficient by a factor of 2 (increased and decreased) did not result in significant changes in COS values (<0.2)."

**4. I don't understand why the acetoclastic methanogenesis was absent here. Generally acetoclastic methanogenesis dominates in lake sediment and hydrogenotrophic methanogenesis in sea sediment. The two pathways generate different d13C-CH4 and d13-DIC pattern. Diffusion and birrigation will also change this pattern. The authors should prove it.**

Here, we clearly show that methanogenesis is 100% hydrogenotrophic at both study sites. As stated L. 284-287:

"hydrogenotrophy becomes an increasingly important $CH_4$ production pathway: i) when labile OM is depleted (Chasar et al., 2000; Hornibrook et al., 2000; Whiticar et al., 1986), ii) with increasing sediment/soil depth (Conrad et al., 2009; Hornibrook et al., 1997), or iii) with decreasing rates of primary production in aquatic environments (Galand et al., 2010; Wand et al., 2006)"

We also replaced l. 269-279:

"Modeled $\delta^{13}C$ profiles were considered acceptable only when they fell within one standard deviation of the measured $\delta^{13}C$ profiles (grey area fills in Fig. 4). Acceptable modeled $\delta^{13}C$ profiles were

obtained only when methanogenesis was 100% hydrogenotrophic, i.e., when $R_3 = 0$ (see section S2.2.2.1)."

**5. The chemical composition of individual molecules in OM pools can be detected from various state-of-the-art instrumentation including GC-MS, LC-MS/MS, HPLC-MS, NMR, Orbitrap MS, and Fourier transform ion cyclotron resonance (FTICR-MS). By combining a suite of previously developed thermodynamic theories ( Kleerebezem and Van Loosdrecht, 2010; LaRowe and Van Cappellen, 2011), one can calculate COS. If the results are consistent, the paper method is more convincing.**

Agreed, presenting data from state-of-the-art analytical methods on the composition of organic molecules could have been convincing, if we were able to isolate the compounds of interest. Indeed, the fraction of the organic C that is degraded in the sediment only represents <5% of the total Corg deposited. It could be challenging to isolate the compounds of interest that we believe are undergoing fermentation. This could well be the subject of a future study.

Nonetheless, fatty acids and alcohols, which are believe to be at the origin of methanogenesis here, are widespread compounds in lake sediments, and are major component of plant organic material (Cranwell, 1981; Matsumoto, 1989).

To better appreciate the point that only a unsignifiant fraction of Corg was degraded in the sediment we added the following text in section 4.1:

"Considering the sediment accumulation rate and sediment $C_{org}$ content given in section 2.1, we calculate an average accumulation rate of $C_{org}$ of $4.7 \times 10^{-11}$ to $1.0 \times 10^{-10}$ and $2.9 \times 10^{-11}$ to $7.6 \times 10^{-10}$ mol C cm$^{-2}$ s$^{-1}$ for lakes Tantaré Basin A and Bédard, respectively. Hence, the total sediment OM degradation rate ($\Sigma R_1 + \Sigma R_2 + \Sigma R_6$) of $1.3 \times 10^{-12}$ and $1.4 \times 10^{-12}$ reported in this study for lakes Tantaré Basin A and Bédard, respectively, would involve only 1.2−2.8% and 0.2−4.8% of the total $C_{org}$ deposited. Given that the remaining 95.2−99.8% of the deposited $C_{org}$ is preserved in the sediment, it is not surprising that the sediment $C_{org}$ concentration is constant with depth (Fig. 2)."

**6. d13C-CH4 in Lake Tantaré Basin A (Fig.3) is very negative (−107.0). Is there some explanation?**

A common explanation given in the literature is the intertwined hydrogenotrophy and methanotrophy. This process is also shown here to produce this local 13C depletion. See L. 305-317.

*REFs:*

Botrel, M., Gregory-Eaves, I. & Maranger, R. Defining drivers of nitrogen stable isotopes (δ15N) of surface sediments in temperate lakes. Journal of Paleolimnology 52, 419-433 (2014).

Clayer, F., Gobeil, C. and Tessier, A.: Rates and pathways of sedimentary organic matter mineralization in two basins of a boreal lake: Emphasis on methanogenesis and methanotrophy: Methane cycling in boreal lake sediments, Limnology and Oceanography, 61(S1), S131–S149, doi:10.1002/lno.10323, 2016.

Conrad, R. (1999). Contribution of hydrogen to methane production and control of hydrogen concentrations in methanogenic soils and sediments. FEMS Microbiology Ecology, 28(3), 193–202. https://doi.org/10.1111/j.1574-6941.1999.tb00575.x

Couture, R.-M., Gobeil, C. and Tessier, A.: Chronology of Atmospheric Deposition of Arsenic Inferred from Reconstructed Sedimentary Records, Environ. Sci. Technol., 42(17), 6508–6513, doi:10.1021/es800818j, 2008.

Cranwell P. A. (1981) Diagenesis of free and bound lipids in terrestrial detritus deposited in a lacustrine sediment. Org. Geochem. 3, 79–89.

Francioso, O., Montecchio, D., Gioacchini, P., and Ciavatta, C. Thermal analysis (TG–DTA) and isotopic characterization (13C–15N) of humic acids from different origins (2005) Applied Geochemistry 20(3), 537-544

Matsumoto G. I. (1989) Biogeochemical study of organic substances in Antarctic lakes. Hydrobiologia 172, 265–289.

---

## Author Response (AR2)

Author's response to reviewer's comments

**Comment on "Mineralization of organic matter in boreal lake sediments: Rates, pathways and nature of the fermenting substrates" by François Clayer et al.**

*Bold and Italic line numbers* refer to lines in the annotated MS (attached below).

Reviewer 2:

**Anonymous Referee #2 report submitted on: 9 July 2020**

**The revised manuscript is improved in many respects. While it still reads somewhat heavy, it is now easier to follow in the aspects that relate to its central theme, methanogenesis. I still have a problem with the sulfur part of the story, however.**

We are grateful to the reviewer for his/her comprehensive and constructive comments. We particularly appreciate the time invested to read through the paper by Couture et al. 2016 to provide some deeply comprehensive feedback.

We believe that the newly revised version of the manuscript does not include any confusion regarding the sulfur cycling since we removed the text that was confusing, provide an alternative explanation for increase SO4 concentration with depth, as suggested by the reviewer, and clarified some aspects regarding our modelling approach.

**Regarding the dissolution of iron sulfides, the authors refer to Couture et al. 2016. That study shows that FeS dissolves in Basic B of Lake Tantare, while in Basin A the dissolution rate is significantly smaller (their Fig. 7) According to Couture et al. 2016, while FeS (as an unstable AVS fraction) dissolves, FeS2 is formed instead. (Curiously, neither of their modeled sulfate profiles actually reproduced the increasing sulfate trend. The dominant pool of sulfur in these sediments was inferred to be in organic form, which is not even considered here but can be potentially significant and can even generate sulfate through mineralization; e.g. Fakhraee et al. 2017)**

**This study, in contrast, claims not only that the FeS dissolution is highly active in Basin A, but also that the precipitation of stable sulfides can be ignored. This is problematic even for the internal logic of the paper: If pyrite precipitation is ignored because iron sulfides are dissolving in these sediments, what is the basis for using the reaction r8 with an end product that cannot form?**

**And yet, this still does not explain the accumulation of sulfate, as the dissolution of sulfides generates only the reduced form, sulfide. Line 238 states: sulfide oxidation by iron oxides is a source of sulfate (r8). This reaction is considered to generate pyrite and sulfate. This is not an actual reaction, however. Oxidation of sulfide produces elemental sulfur. A quarter of it may be converted to sulfate via disproportionation (with the other three quarters regenerating sulfide). But sulfate could hardly be expected to accumulate in the anoxic sediment, as sulfate reduction is a much more efficient sink for sulfate than disproportionation is a source.**

Agreed, interpreting the SO4 profiles reported in our manuscript is not straightforward.

To simplify this aspect of the manuscript, we now acknowledge the fact that explaining the sediment S cycling is out of the scope of our study and only mention the mineralization of OM as a potential source of SO4 given the importance of organic S in the sediment OM pool (see *L. 253-258* and our response to your last comment).

Regarding the increasing SO4 trend below 5 cm depth, note that Couture et al. 2016 report measured and modeled porewater SO4 profiles only down to 9.5 cm depth. The limited depth range prevented them to detect any significant trend in SO4 concentrations. In contrast, the profiles reported in our manuscript display porewater SO4 concentrations down to 23 to 25 cm depth.

**The sulfate profiles are claimed to be successfully reproduced with the model that is based on Clayer et al. 2016. The description of the model in Clayer et al. 2016, however, is incomplete and does not specify the reactions in the sulfur cycle. So while I tried to understand the processes that could lead to sulfate accumulation, I could not identify the specific reactions on which the simulation was based. The only reaction listed in Clayer et al. 2016 that produces sulfate is the aerobic oxidation of sulfide, which obviously cannot operate below the depth of oxygen penetration.**

Please, note that the modelling approach described here (and in Clayer et al., 2016) is not based on any specific reaction. A simple mass-conservation equation (Eq. 2) is solved for the net reaction rate to reproduce the concentration profile of each solute, separately. In other words, the fluxes of a given solute caused by diffusion, bioirrigation and net reaction are balanced to reproduce the measured concentration profile.

This procedure, often referred to as "inverse modelling", yields independent net reaction rate for each solute. The reactions presented in section 2.5 are not included in the modelling, we use them to interpret the net reaction rates obtained for each solute by inverse modelling, and to constrain the effective reaction rates (as described in Section 2.6).

It is thus straightforward to successfully reproduce the measured concentration profiles with inverse modelling since the depth distribution of the net SO4 reaction rate are fitted to the measured concentration profile.

The modelling approach presented by Couture et al. 2016 is a forward modelling approach that links the various solutes through reactions and kinetics expressions. This is not the case here.

To ensure that our phrasing do not lead to any misunderstanding regarding our modelling approach, we performed the following modifications:

*L. 131-133*

"The following one-dimensional mass-conservation equation (Boudreau, 1997):

$$\frac{\partial}{\partial x}\left(\varphi D_s \frac{\partial[\text{solute}]}{\partial x}\right) + \varphi\alpha_{\text{Irrigation}}([\text{solute}]_{\text{tube}} - [\text{solute}]) + R_{\text{net}}^{\text{solute}} = 0 \qquad (2)$$

was used to separately model the porewater profile of each relevant solute, i.e., $CH_4$, DIC, $O_2$, Fe and $SO_4^{2-}$, considering steady state and negligible solute transport by bioturbation and advection."

We also renamed section 2.4 for more clarity and include the term "inverse modelling" as follows:

*L. 127*

"**2.4 Inverse modeling of porewater solutes**"

And finally, the sentences regarding the calculations of solute activities and saturation index with WHAM (originally at **L. 128-130**) were moved at the end of section 2.4 (now at **L. 155-157**), given their secondary importance.

**Thus the puzzle of the sulfur cycle remains. It seems to me, however, that this cycle is invoked in the paper mainly to justify the source of H2 (line 610). But there are other ways to generate hydrogen, as already acknowledged in the paper. Perhaps the authors might consider making their argument without an explicit consideration of the cryptic sulfur cycling, or otherwise they need to describe it in more realistic detail.**

The reviewer's suggestion to remove the explicit inclusion of the cryptic sulfur cycling appeared as the most suitable solution. Indeed, it removes our ambiguous interpretations of the SO4 profiles, and allows to focus the manuscript on methanogenesis and fermentation.

In addition, we now mention SO4 generation through OM mineralization as a possible explanation of the SO4 increase with depth, as suggested above by the reviewer.

Consequently, we removed reaction r8 from Table 1 and its description in section 2.5 (**L. 168-173**). We also streamlined the description of the SO4 porewater profiles in section 3.1 and simplified the beginning of section 4.2.

**L. 253-258** now read:

[revised manuscript text omitted]